# Source-sector contributions to European ozone and fine PM in 2010 using AQMEII modeling data

Prakash Karamchandani [1], Yoann Long[2], Guido Pirovano[3], Alessandra Balzarini[3], Greg Yarwood[1]

[1]Ramboll Environ, 773 San Marin Drive, Suite 2115, Novato, CA 94998, USA
[2]Ramboll Environ, Immeuble Le Cézanne, 155 rue de Broglie, 13100 Aix-en-Provence, France
[3]Ricerca sul Sistema Energetico (RSE SpA), Via Rubattino, 54 - 20134 Milano, Italy

*Correspondence to*: Prakash Karamchandani (pkaramchandani@ramboll.com)

**Abstract.** Source apportionment modeling provides valuable information on the contributions of different source sectors and/or source regions to ozone ($O_3$) or fine particulate matter ($PM_{2.5}$) concentrations. This information can be useful in
designing air quality management strategies and in understanding the potential benefits of reducing emissions from a particular source category. The Comprehensive Air quality Model with Extensions (CAMx) offers unique source attribution tools, called the Ozone and Particulate Source Apportionment Technology (OSAT/PSAT), which track source contributions. We present results from a CAMx source attribution modeling study for a summer month and a winter month using a recently evaluated European CAMx modeling database developed for Phase 3 of the Air Quality Model Evaluation International
Initiative (AQMEII). The contributions of several source sectors (including model boundary conditions of chemical species representing transport of emissions from outside the modeling domain as well as initial conditions of these species) to $O_3$ or $PM_{2.5}$ concentrations in Europe were calculated using OSAT and PSAT, respectively. A one week spin-up period was used to reduce the influence of initial conditions. Evaluation focused on 16 major cities and on identifying source sectors that contributed above 5%. Boundary conditions have a large impact on summer and winter ozone in Europe and on summer
$PM_{2.5}$, but are only a minor contributor to winter $PM_{2.5}$. Biogenic emissions are important for summer ozone and $PM_{2.5}$. The important anthropogenic sectors for summer ozone are transportation (both on-road and non-road), energy production and conversion, and the industry sector. In two of the 16 cities, solvent and product also contributed above 5% to summertime ozone. For summertime $PM_{2.5}$, the important anthropogenic source sectors are the energy sector, transportation, industry, and agriculture. Residential wood combustion is an important anthropogenic sector in winter for $PM_{2.5}$ over most of Europe,
with larger contributions in central and eastern Europe and the Nordic cities. Other anthropogenic sectors with large contributions to wintertime $PM_{2.5}$ include energy, transportation, and agriculture.

## 1 Introduction

Photochemical grid models (PGMs), such as the Comprehensive Air quality Model with Extensions (CAMx; Ramboll Environ, 2014) and the Community Multiscale Air Quality (CMAQ) model (Byun and Schere, 2006), are widely used in air

quality management to assess the effectiveness of potential control strategies for secondary pollutants such as $O_3$ and $PM_{2.5}$. Source apportionment (SA) analysis is an important component of this process to identify source sectors and/or regions that are contributing to $O_3$ and $PM_{2.5}$. The traditional approach to source attribution analysis is "brute-force" or "zero-out" sensitivity analysis in which the emissions from a given source sector are removed to quantify the contribution of that sector.

This approach is expensive and impractical for assessing the contributions of a large number of source categories and suffers from the limitation that the sum of zero-out impacts over all sources will not equal the total concentration (Koo et al., 2009). Tagged species methods, such as the Ozone and Particulate Source Apportionment Technology (OSAT/PSAT) in CAMx (Dunker et al., 2002; Yarwood et al., 2007), can efficiently track contributions from many source sectors and/or regions and provide source contributions that sum to the total concentration. These methods are increasingly being used to help

understand complex air quality issues (e.g., Wagstrom et al., 2008; Burr and Zhang, 2011;Baker and Kelly, 2014; Collet et al., 2014; Wang et al., 2009; Li et al., 2012; Skyllakou et al., 2014).

Many source attribution studies in Europe have used receptor models and back trajectory analysis for inert pollutants, i.e., primary particulate matter (e.g., Querol et al., 2001; 2004; 2009; Belis et al., 2011; 2013; Viana et al., 2014). Source

attribution studies in Europe for secondary pollutants, such as ozone and secondary $PM_{2.5}$, have used PGMs with the zero-out approach and Skyllakou et al. (2014) used the CAMx PSAT approach to distinguish the contributions of local and regional sources to fine PM in Paris. Reis et al. (2000) studied the impact of road traffic emissions on regional ozone levels in Europe by zeroing-out traffic emissions. Derwent et al. (2005) used a similar approach to determine the contribution of shipping emissions to ozone and acid deposition in Europe. Sartelet et al. (2012) estimated the contributions of biogenic and

anthropogenic emissions to $O_3$ and PM concentrations in Europe and North America by zeroing-out one source category at a time. Under the TRANSPHORM (Transport related Air Pollution and Health Impacts-Integrated Methodologies for Assessing Particulate Matter) program, the source contributions of transport emissions (road transport, shipping, aviation) to $O_3$ and PM contributions in Europe were assessed using WRF-CMAQ and the zero-out approach (TRANSPHORM, 2014). Derwent et al. (2008) conducted sensitivity studies with a global chemical transport model to understand the effects of long-

range transport from North America and Asia to surface ozone in Europe. Aksoyuglu et al. (2016) estimated the contribution of ship emissions to the concentrations and deposition of air pollutants in Europe using CAMx with a zero-out approach.

All of the above PGM source attribution studies for Europe investigated the contributions of a limited number of source categories. Examples of European source attribution studies addressing a larger number of source categories include EMEP

(2009), Brandt et al. (2013) and Tagaris et al. (2015). The European Monitoring and Evaluation Programme (EMEP) study used 15% sector-by-sector emission reductions in the Unified EMEP model to determine contributions from different emission sectors to depositions and air concentrations in countries within the EMEP domain for 2006 (EMEP, 2009). Brandt et al. (2013) used the EVA model system, which includes the Danish Eulerian Hemispheric Model (DEHM), with a tagging

approach to evaluate the contributions from the 10 Standard Nomenclature for Air Pollution (SNAP) source sectors to air pollution related health costs in Europe and Denmark for the 2000 base year. More recently, Tagaris et al. (2015) used the zero-out approach with CMAQ to calculate the contributions of emissions from the 10 SNAP source sectors to air quality over Europe for a summer month (July 2006).

The study described in this paper differs from and complements the previous source attribution studies. We take advantage of the source apportionment tools (OSAT and PSAT) in CAMx to calculate the contributions of the 10 SNAP sectors, biogenic emissions, dust emissions, and sources outside the modeling domain (boundary conditions) to a large number of cities in Europe. The CAMx modeling database for 2010 developed as part of the Air Quality Model Evaluation International Initiative (AQMEII) is used for the analysis.

AQMEII is a joint regional air quality model evaluation effort between the North American and European modeling communities to improve the understanding of uncertainties in model predictions of ozone and $PM_{2.5}$, and to use this knowledge to guide model improvements (Rao et al., 2011). In Phase 1 of the AQMEII, a large number of offline photochemical air quality models were applied and evaluated in Europe and North America for the year 2006 using consistent inputs to the extent possible (Rao et al., 2011; Galmarini et al., 2012). The second phase of AQMEII was dedicated to the evaluation of online coupled chemistry-meteorology models over both continents with a primary focus on the year 2010 (Galmarini et al., 2015). In Phase 3, the focus is on diagnostic evaluation through sensitivity studies on model inputs and spectral decomposition of model errors (Galmarini et al., 2016; Solazzo and Galmarini, 2016; Solazzo et al., 2016), as well as on collaboration with the Task Force on Hemispheric Transport of Air Pollution (TF-HTAP) on coordinating global/hemispheric/regional modeling experiments to characterize long-range transport. As part of Phase 3, a CAMx modeling database has been developed and evaluated for Europe (Solazzo et al., 2016) and this database was used in the study described in this paper.

## 2 Model Setup

### 2.1 Model configuration, domain, and inputs

Solazzo et al. (2016) developed a 2010 CAMx database for Europe and applied CAMx version 6.1 (Ramboll Environ, 2014) over the European Union (EU). The simulations used the Carbon Bond 2005 (CB05) gas phase chemistry (Yarwood et al., 2005) and the Coarse-Fine (CF) aerosol scheme. CAMx was applied for the whole year for a domain covering Europe and a portion of Africa. The domain (see Fig. 1) is defined in a Lambert Conic Conformal projection that includes 270 by 225 grid cells with a 23 km horizontal grid resolution.

Input meteorological data were generated using WRF-Chem 3.4.1, the coupled chemistry version of the Weather Research and Forecasting (WRF) Model (Skamarock et al., 2008), driven by the European Centre for Medium-Range Weather Forecasts (ECMWF) analysis fields. WRF-Chem was used rather than WRF to obtain emission estimates for wind-blown dust. Analysis nudging for wind speed, temperature and water vapor mixing ratio was employed both within and above the boundary layer, with a nudging coefficient of 0.0003 sec$^{-1}$. The WRF vertical grid with 33 layers extends from the surface to a fixed pressure of 50 hPa (about 20 km), with a surface layer depth of 24 m. The WRFCAMx pre-processor (version 4.2; Ramboll Environ, 2014) was used to create CAMx input files collapsing the 33 layers used by WRF to 14 layers in CAMx but keeping layers up to 230 m above ground level identical to the WRF layers.

Anthropogenic emissions for calendar year 2009 were derived from the TNO-MACC_II emission inventory (Kuenen et al., 2014; Pouliot et al., 2015) resolved by SNAP sector (see Table 1). The primary data sources were national emission inventories developed by European countries in accordance with guidance provided by the European Environment Agency (EEA, 2013). SNAP sector 34 combines "industrial combustion" (sector 3) with "industrial processes" (sector 4) to mitigate inconsistent classification of sources to sector 3 or 4, as discussed by Kuenen et al. (2014). Supplemental Section A provides a summary of $SO_2$, NOx and $PM_{2.5}$ emissions from the 9 SNAP sectors for the summer and winter months and presents spatial maps of total surface emissions and surface emissions for some sectors.

For the road transport sector (SNAP 7), it should be noted that the provided emission inventory does not include information on the composition of the vehicle fleet in different cities in Europe because the emission inventory was made available to the AQME participants with source contributions grouped according the SNAP classification, but without any additional information about the car fleet or other proxies introduced in emission computation. However, the MACC-II emission inventory (Kuenen et al., 2014) that was used for this study was constructed by using the reported emission national totals by sector. Therefore, for each country a representative car fleet was used and the differences in fleet composition among different countries are implicitly accounted for in the provided emission inventory. The non-road transport sector (SNAP 8) includes a variety of emission sources, such as off-road transport (shipping, railways, aviation, inland waterways) and machinery (agriculture, forestry, industry, military, airports).

Biogenic VOC emissions were estimated by applying the Model of Emissions of Gases and Aerosols from Nature (MEGAN; Guenther et al., 2012) v2.04. Sea salt emissions were estimated using published algorithms (de Leeuw et al., 2000; Gong, 2003). Dust emissions were based on the GOCART (Ginoux et al., 2001; 2004) model implemented in WRF-Chem (Zhao et al., 2010). Chemical boundary conditions were derived from the Monitoring Atmospheric Composition and Climate (MACC) project using the Composition–Integrated Forecast System (C-IFS) model (Flemming et al., 2015). The MACC data were available at 3-hourly time intervals with horizontal resolution of 1.125 x 1.125 degrees. Variables were provided

as 3D fields in pressure hybrid vertical coordinates and included gas phase species, namely CO, $O_3$, NO, $NO_2$, PAN, $HNO_3$, $CH_2O$ (formaldehyde), $SO_2$, $H_2O_2$, $C_2H_6$ (ethane), $CH_3COCH_3$ (acetone), $CH_3OH$ (methanol), $C_3H_8$ (propane), $C_2H_5OH$ (ethanol), $C_2H_4$ (ethene), PAR (paraffins), ALD2 (aldehydes), OLE (olefins), $C_5H_8$ (isoprene), CHOOH (formic acid), $CH_3OOH$ (methylperoxide), ONIT (organic nitrates), and aerosol species (dust, sulfate, hydrophobic and hydrophilic

organic matter, and hydrophobic and hydrophilic black carbon). Mineral dust aerosols were provided in three different size bins, ranging from 0.03 to 20 µm. More information on MACC data as well as their evaluation against a set of ground-based measurements can be found in Inness et al. (2013) and Giordano et al. (2015).

## 2.2 Model performance evaluation summary

Solazzo et al. (2016) conducted a detailed model performance evaluation of CAMx for 2010 in the framework of the

AQMEII Phase 3 project. Here we present a brief summary of model performance using a set of ground-based observations belonging to the Airbase network (http://www.eea.europa.eu/data-and-maps/data/airbase-the-european-air-quality-database-8) and covering most of the computational domain. Only background stations are considered in the analysis. Furthermore, analysis was carried out for the whole set, including Urban, Suburban and Rural sites (All Background, AB), as well as for Rural Background sites only (RB).. Note that the Rural Background sites are the most appropriate for model performance

evaluation because the coarse resolution (23 km) used in the simulation cannot reproduce local scale processes that can take place, particularly within the urban areas, and that can also influence the observed values at background sites. However, since urban areas were the main focus of the source attribution analysis presented here, the evaluation was performed over background stations, not just the Rural Background sites. The comparison of the model performance for these two different sets of sites provides a quantitative evaluation of the possible degradation of the CAMx results when evaluated at urban sites,

whose spatial representativeness is not always adequate to the model resolution, even for background sites. As noted by Terrenoire et al. (2015), model performance for RB sites is better than for UB sites even when using a fine resolution of 8 km.

The model performance was evaluated over the whole year and based on daily mean concentrations. Only stations that had

more than 75% of data availability on a yearly basis have been included in the comparisons. The number of available stations ranges according to the chemical species. The highest availability was noted for $NO_2$, with more than 2500 stations. Ozone and $SO_2$ were available at more than 1500 sites, while for $PM_{10}$ more than 2300 sites were available. $PM_{2.5}$ observations were available in about 700 sites over all Europe, with about 300 sites corresponding to RB stations. Model performance was evaluated also at city level for selected cities for the winter (January to March) and summer (July to

September) seasons of 2010 for consistency with the source attribution analyses described in this paper.

## 2.2.1 Annual model performance evaluation for entire domain

Table 2 provides a domain wide summary of model performance for the AB and RB sets of stations. The statistical performance measures used in the evaluation are defined in Supplemental Section B (Sect. B1). "Correlation" refers to the Pearson correlation coefficient ("r") and expresses the temporal correlation between the observed and computed daily mean concentrations. The yearly mean of the observed $SO_2$ concentration is 2.3 ppb, while the modeled value is 1.2, corresponding to a 48 % low bias. Similar results are noted at RB sites (NMB = -41%). $NO_2$ yearly mean concentrations are clearly underestimated when all background sites are considered. However, when the analysis is limited only to RB sites, which are more suitable for comparisons with a model run using a 23 km horizontal grid resolution, there is a noticeable improvement in model performance. The NMB improves from -56.3 % to -29.4% and the NME decreases from 60% to 47.8%. As a consequence, the RMSE drops from 11.7 to 5.4 ppb, and the daily correlation grows from 0.52 to 0.60. The under-estimation of $NO_2$ concentrations may be because the grid resolution is too coarse to resolve many of the monitoring locations, or alternatively indicates that NOx emissions are under-estimated in this inventory.

Annual mean ozone concentrations at AB sites are overestimated (NMB = 21.1%), while the standard deviation (SD) of the yearly series is correctly reproduced (standard deviation of 11.3 ppb observed versus 12 ppb modeled). Similar performance for SD is observed at RB sites, together with a clear improvement in terms of the yearly mean, as pointed out by the decrease of NMB and NME. These results suggest that the ozone bias at AB sites is partially due to overestimation at urban and suburban sites, where the horizontal grid resolution is insufficient to resolve ozone suppression at monitors by nearby sources of NOx. This hypothesis is confirmed by analysis of the Ox concentration (Ox = $O_3$+$NO_2$) that removes the local effect of NOx titration. Ox concentrations at both AB and RB sites are very well reproduced in terms of both mean and SD. Also, the temporal variation of Ox concentrations is well reproduced as shown by the correlation value (0.64 and 0.67 at AB and RB sites, respectively).

$PM_{10}$ concentrations are underestimated at AB sites (NMB = -19%), but the bias for RB sites is small (NMB = 0.4% and FB = 3.5%). Conversely, the NME (and FE) remains high for both sets of stations. In particular, the NME increases from 51.7% at AB sites to 55% at RB sites. The temporal correlations are also low (< 0.3) in both cases. The overall performance suggests that CAMx correctly captured the yearly mean burden of aerosol but not its temporal evolution. This could be due to compensating errors among different sources that could be either underestimated or overestimated. The correlations for $PM_{2.5}$ are better (correlation = 0.48 and 0.52 at AB and RB sites, respectively), although there is more underestimation bias for $PM_{2.5}$ than for $PM_{10}$. These results suggest that coarse PM mass is likely overestimated and its temporal evolution is not correctly reproduced by CAMx. Note that a large fraction of the coarse PM can be attributed to dust and/or sea salt sources and there are large uncertainties in estimating emissions from these sources.

Supplemental Section B provides additional details and discussion on the spatial and temporal annual performance of the model. Below, we present model performance results for specific cities during the winter and summer seasons.

### 2.2.2 Seasonal model performance evaluation for selected cities

Model performance was also evaluated at those cities selected for source apportionment analysis (see Sect. 2.3 for the selected cities) that had available background stations. For each city, all Airbase background stations belonging to an area of about 50 x 50 km$^2$ placed around the city were selected. For all cities, at least two sites were included when available. The analysis was carried out over two quarters: January-March and July-September 2010. The two quarters cover the months of February and August that were selected for SA analysis. Ozone was evaluated only for the summer season, while PM$_{2.5}$ was
evaluated for both periods. MPE was based on the same indicators used for the performance analysis of the annual results (see Sect. B1 in Supplement B).

The performance evaluation results for summer ozone are summarised in Table 3 and Figures 2 and 3. CAMx reproduced the ozone summer mean fairly well, though showing a slight and systematic overestimation. FB was lower than 20% at all sites,
except for two cities in Eastern Europe, Warsaw (24%) and Budapest (35%). Temporal correlations ranged between 0.6 and 0.8 for all sites, with the exception of two Mediterranean sites, Barcelona and Athens, where the correlation dropped to 0.4. As shown in Figure 3, the degradation in correlation in Barcelona is due to an overestimation taking place in July and September, while in August ozone concentrations were correctly captured. In contrast, the worsening in model performance at Athens is probably due to the higher variability, in both space and time, shown by observed data, which is not captured by
CAMx.

The analysis of CAMx time series of ozone concentrations illustrates the systematic overestimation in Budapest for all percentiles, while in Warsaw the overestimation is primarily associated with the median and the third quartile. At all other sites, CAMx is able to capture both the seasonal trend, slightly decreasing from July to September, as well as the
development of most of the short term episodes (e.g. during the first and second half of July in Lisbon, Paris, Berlin, Amsterdam and London).

PM$_{2.5}$ was evaluated for both summer and winter seasons. During the warm season, the observed mean concentration ranges between 6 and 14 µg/m$^3$ (see Table 4 and Figure 4), except in Athens and Warsaw, where the seasonal values are around 25
and 20 µg/m$^3$, respectively. PM$_{2.5}$ mean concentrations were underestimated at most sites, with FB substantially close to 0 in Copenhagen and London, and ranging between -5% and -20% in Barcelona, Lisbon, Berlin, Oslo and Helsinki. As already mentioned, the worst performance was for Athens and Warsaw. Finally, PM$_{2.5}$ concentrations are partially overestimated in

Paris and Amsterdam. FE is generally lower than 40%, again with the exception of Athens and Warsaw proving that, beyond the mean values, the whole distribution of the daily mean concentrations is also reproduced fairly well. Conversely, temporal correlations range between 0.2 and 0.6, pointing out the model difficulty in capturing the exact temporal variability of the observed values. As shown in Figure 5, this is probably due to the very small variability of the observed concentrations over

the summer season. At the Lisbon site, CAMx strongly overestimates an episode, showing a concentration about twice the observed one, which is probably due to a corresponding overestimation of the contribution of the natural sources (e.g., sea salt).

During the winter season, CAMx is partially able to capture the spatial variability of the observed concentrations, which

range between 9 µg/m$^3$ (Lisbon) and 42 µg/m$^3$ (Warsaw), as shown in Table 5 and Figure 6. CAMx clearly underestimates the seasonal mean concentration in Warsaw (FB=-40%) and Berlin (FB=-20%), which show the highest observed values, as well as in Athens and Barcelona (FB =-36%). Discrepancies between modeled and measured values in Eastern Europe are mainly related to the underestimation of several very strong episodes taking place in January over the area (Figure 7). However, CAMx performs better in February (the month used for the SA analysis) and March. CAMx correctly reproduces

the mean concentrations in Central-Western Europe (London, Paris and Amsterdam), while it partially overestimates the observations at Northern Europe sites. The latter is probably due to an overestimation of the sea salt contribution.

The seasonal analysis performed at city level shows that CAMx is generally able to capture the spatial and temporal patterns of the pollutant concentrations across Europe providing confidence in the different source contributions estimated at each

city, discussed in the following section. Moreover, in the case of PM$_{2.5}$, CAMx is also able to correctly capture the seasonal variations.

## 2.3 Source attribution modeling

The source attribution modeling with CAMx used the OSAT and PSAT tagged species methods in CAMx version 6.1 (Ramboll Environ, 2014). In addition to the SNAP sector emission categories, the contributions of biogenic emissions, dust

and sea salt emissions (for PM), and boundary conditions of model chemical species were explicitly tracked. Secondary organic aerosol (SOA) was not apportioned by PSAT because of the high computer memory requirement to track SOA categories on the large CAMx modeling grid. The total biogenic and anthropogenic SOA were both available from the CAMx CF aerosol scheme. Note that the PSAT approach apportions contributions to PM species independently, and can thus handle particulate ammonium having different source contributions (e.g., from agriculture) than particulate nitrate (e.g.,

from urban traffic emissions).

The source attribution simulations were conducted for a summer month (August 2010) and a winter month (February 2010). Although a model spin-up period of one week (last week of January 2010 for the winter simulation and the last week of July 2010 for the summer simulation) was used to minimize the influence of initial conditions, the contributions of initial conditions to surface ozone and PM concentrations are included in the boundary condition attribution component in the discussion of results in the following section.

## 3 Results

We selected 16 cities, representing the Nordic countries, countries in western, central, and eastern Europe, and countries along the Mediterranean coastline, for the source attribution analysis. The contributions of the various source sectors to ozone and $PM_{2.5}$ concentrations were calculated for these cities and are discussed in this section. The calculations were conducted using horizontal bilinear interpolation over 8 grid cells around each city location.

### 3.1 Ozone source apportionment-summer

The European standard for ozone is based on the maximum daily 8 hour mean (not to exceed a threshold of 120 µg/m3, about 60 ppb, for 25 days averaged over 3 years). Accordingly, the source apportionment results for ozone are presented for the maximum daily 8 hour average (referred to as H1MDA8) for the summer month. Ozone results for the winter month are not discussed here because H1MDA8 levels at all the selected cities are less than the threshold and because boundary conditions dominate the ozone levels in winter, with contributions at the 16 cities ranging from a low of 61% to a high of 96%. The spatial pattern of calculated H1MDA8 ozone concentrations across the modeling domain is shown in Figure C1 in Supplement C. Over most of western and northern Europe, ozone levels are below 60 ppb. The highest ozone values (about 120 ppb) are predicted near Moscow, Russia. The 60 ppb level is exceeded in some of the Mediterranean countries and in parts of central and eastern Europe.

The source attribution results for summertime H1MDA8 ozone in each city are reported in Table 6 for contributors of 5% or more. In the 4 cities near the Mediterranean from Lisbon, Portugal in the west to Istanbul, Turkey in the east, H1MDA8 ozone in August 2010 is estimated to be above or very close to the standard of 60 ppb. Boundary conditions are the largest contributor to H1MDA8 ozone in all 4 Mediterranean cities, with contributions ranging from 26% to 34% from east (Istanbul, Athens) to west (Barcelona, Lisbon). Contributions from on-road transport (SNAP 7) are the next largest (20% to 24%) at 3 of the 4 cities (Lisbon, Barcelona, Athens). At Istanbul, the second highest contribution (24%) comes from biogenic emissions, while on-road transport is the third largest contributor at 15%. Non-road transport (SNAP 8; 12% to 18% contribution) and biogenic emissions (15% to 24% contribution) are also significant contributors at all 4 locations. The

other anthropogenic sectors contributing 5% or more to summertime ozone in the Mediterranean cities are industry (SNAP 34; 6 to 8% contribution) and the energy sector (SNAP 1; 5 to 8% contribution).

Boundary conditions are again an important contributor in the 4 cities in central and eastern Europe making the largest
contribution in Minsk (25%) and Warsaw (28%) and the second largest contribution in Budapest (29%) and Kiev (21%). Road transport is the largest contributor (35%) in Budapest, while biogenic emissions are the largest contributor (33%) in Kiev. Road transport is the second largest contributor in Warsaw (25%) and biogenic emissions are the second largest contributor (23%) in Minsk. Biogenic emissions contribute less in Budapest (10%) and Warsaw (14%) than in the other two cities. Other important contributing source sectors in the central and eastern European cities are the energy sector (9% in
Kiev to 17% in Warsaw), the non-road sector (7% to 10% contribution) and the industry sector (5% to 7% contribution).

In all 4 cities in western Europe (Paris, London, Amsterdam, and Berlin), H1MDA8 ozone concentrations are below the 60 ppb threshold. Boundary conditions are the largest contributor (29% to 59%) to HD1MA8 ozone at all 4 cities, and contribute more than half the HD1MA8 ozone in London and Paris and nearly 50% in Berlin. Road transport is the next
largest contributor in Paris (13%) and Berlin (17%), while non-road transport (12%) and biogenic emissions (21%) are the second largest contributors in London and Amsterdam, respectively. For London, the most relevant non-road transport emissions are likely due to the very intense shipping activity along the Channel resulting in large NOx emissions (see, for example, Figure 8 in Kuenen et al., 2014). Road transport contributions in London and Amsterdam rank third at 11% and 19%, respectively. Non-road transport contributes less than 10% to HD1MA8 ozone in Paris, Amsterdam and Berlin. The
energy sector is an important contributor (13%) in Berlin, and contributes 5% to 6% in London and Amsterdam. The energy sector contribution in Paris is small (less than 3%), since France derives over 75% of its electricity from nuclear energy. The solvent and product use sector (SNAP 6) contributes 6% and 10% to summertime ozone in Paris and Amsterdam, respectively, but its contributions in London and Berlin are less than 3%.

Like the selected cities in western Europe, H1MDA8 ozone levels in the Nordic cities (Oslo, Copenhagen, Stockholm, Helsinki) are below the European threshold of 60 ppb. Boundary conditions again play an important role for ozone in these cities, and are the largest contributors in 3 of the 4 cities. Road transport is the largest contributor (24%) in Stockholm followed by boundary conditions (21%). Road transport contributions to the other 3 cities range from 12% in Oslo to 23% in Copenhagen. Non-road transport is an important contributing sector (14 to 21%) and its contributions in Oslo and Helsinki
are higher than on-road transport. As noted by Kukkonen et al. (2016), emissions from shipping and harbors are an important non-road transport influence for Oslo and Helsinki. Biogenic emissions are also important contributors in all 4 cities, with contributions ranging from 12 to 20%. The energy sector contributes 12 to 13% in Helsinki, Stockholm and Copenhagen, but its contribution to H1MDA8 ozone in Oslo is slightly less than 5%.

Figure 8 shows the source apportionment results for the 16 cities across the distribution of summertime MDA8 ozone values. Results are shown for the upper and lower quartiles, the median, and the 90[th] percentile value for each city. Boundary conditions are clearly the primary contributors across the MDA8 ozone distribution at a majority of the cities selected for analysis and particularly in London and Paris. Except for Budapest, boundary conditions are the primary contributors at all cities at the low end of the distribution. Road transport contributions are important in many cities, particularly Budapest and Athens (across the distribution), Warsaw, Lisbon, Minsk, Kiev and Berlin (at the higher end of the distribution). Non-road transport contributions are important at the Mediterranean cities and at the Nordic cities, particularly at the higher end of the distribution. Biogenic emissions are the highest contributors in Kiev at the high end of the distribution, and are also important in the Mediterranean cities, and most of the other selected cities. Contributions from the energy sector contributions are important in the central and eastern European countries, particularly Warsaw.

As noted previously, boundary condition contributions also include the contribution of initial conditions, which are expected to decrease over the one week spin-up period and the subsequent month-long simulation period. At the end of the one week spin-up period (i.e., on August 1), initial condition contributions to summertime H1MDA8 ozone were estimated for 6 cities (Paris, Lisbon, Warsaw, Athens, Oslo, and London) and ranged from 5% at Lisbon (H1MDA8 ozone of 38 ppb) to 54% at Oslo (H1MDA8 ozone of 22 ppb).

## 3.2 PM$_{2.5}$ source apportionment-summer

The European standard for fine PM is an annual average concentration of 25 µg/m$^3$. Since we obtained source attribution for only two months, our discussion of the PM$_{2.5}$ source attribution focuses on the summer and winter monthly average concentrations. The spatial patterns of monthly mean PM$_{2.5}$ concentrations for August 2010 across the modeling domain are shown in Figure C2 in Supplement C. The highest PM$_{2.5}$ concentrations are near the southern and southeastern boundaries of the domain and in the Mediterranean countries. These high concentrations are likely due to the transport of Saharan dust from North Africa from the boundary conditions as well as from the dust emissions within the modeling domain (which includes part of the Sahara) estimated by the GOCART model in WRF-Chem. While most of the Saharan dust is coarse, a significant fraction is in the fine mode (e.g., Zauli Sajani et al., 2012; Figure 7 in Pio et al., 2014). Removing the dust component from the calculated total PM$_{2.5}$ concentrations reduces the highest concentrations along the southern boundary by a factor of 2.

Table 7 shows the source attribution results for monthly mean PM$_{2.5}$ concentrations in August 2010. In the Mediterranean cities of Lisbon, Barcelona, Athens and Istanbul, boundary conditions are the largest contributors to mean August PM$_{2.5}$ concentrations, with contributions ranging from 38% to 49%. Non-road transport and SOA are the second and third largest

contributors in Lisbon and Barcelona. In Athens, the energy sector and non-road transport are the second and third largest contributors, while in Istanbul the industry sector is the second largest contributor and SOA and the energy sector are the third largest contributors. Road transport contributions are less than 5% in Istanbul and less than 10% in Lisbon and Athens. The highest on-road transport contribution to the selected Mediterranean cities is 10% in Barcelona. The industry sector contributions in all 4 Mediterranean cities are 5% or more, while the SOA contributions in the 4 cities are 8% or more. The agriculture sector (SNAP 10) contribution to August 2010 mean $PM_{2.5}$ concentrations is 7 to 8% in Athens and Istanbul and less than 5% in the other 2 cities.

Boundary conditions are important contributors to monthly average $PM_{2.5}$ concentrations at cities in central and eastern Europe as well, as shown in Table 7, but the relative BC contributions in these regions are lower than those in southern Europe. BCs are the largest contributors in Minsk and Kiev, while the energy sector is the largest contributor in Budapest and Warsaw. The dominant component (> 60%) of the boundary condition contribution in Minsk and Kiev is primary fine crustal material. The energy sector contributions range from 9% in Kiev to 24% in Warsaw. SOA are also important contributors in all 4 cities and are the second largest contributors in Minsk (18%) and Kiev (17%). The agriculture sector also has a large contribution in all 4 cities (12% to 14%), suggesting that ammonia emissions from agricultural activity leads to formation of particulate nitrate. The industry sector contributes from 6% to 9% of $PM_{2.5}$ concentrations in the 4 cities. Road transport contributions are 8% in Budapest and Minsk and 10% in Warsaw, but less than 5% in Kiev. Non-road transport contributions are more than 5% in the 4 cities, but less than 10%.

Boundary conditions are not large contributors to the August monthly average $PM_{2.5}$ concentrations in any of the 4 western European cities. Boundary condition contributions range from 9% in Amsterdam to 14% in Paris and Berlin. SOA are the largest contributors in London, Paris, and Berlin, while non-road transport is the largest contributor (28%) in Amsterdam. Non-road transport is an important contributor in the other 3 cities as well, with contributions ranging from 14% in Berlin to 23% in London. The energy sector has a 15% contribution in Berlin, but less than 10% in the other 3 cities. Agriculture has a large contribution (14%) in Paris, but lower contributions in Berlin (8%) and Amsterdam (6%). Agriculture contributions to the mean August 2010 $PM_{2.5}$ concentrations in London are less than 5%. Road transport is an important but not a major contributor (12 to 13%) in any of the 4 western European cities.

The source attribution results for the 4 cities in the Nordic countries show the decreasing influence of boundary conditions in the northern portion of the modeling domain. Boundary condition contributions are not as large as for some of the cities to the south and range from 10 to 15%. SOA and non-road transport are the highest contributors in Oslo and contribute about 25% each. SOA is the largest contributor (about 31%) in Helsinki and Stockholm, while non-road transport is the largest contributor in Copenhagen. Non-road transport is the second highest contributor in Stockholm. Energy sector emissions

contribute from 7 to 12% to monthly mean $PM_{2.5}$ concentrations, while the on-road transport sector contributes 9 to 13%. Residential combustion (SNAP 2) contributes 11% in Oslo but less than 5% in the other 3 Nordic cities.

Figure 9 shows the source apportionment results for the 16 cities across the distribution of daily average summertime $PM_{2.5}$
concentrations. In the Mediterranean cities, boundary conditions are more important at the high end of the distribution, suggesting that the higher $PM_{2.5}$ concentrations in these cities are often associated with transport from outside of the domain. Non-road transport is an important contributor across the distribution in Barcelona and Lisbon, but less important in Athens and Istanbul. SOA contributions are important in all cities, particularly Athens.

Boundary conditions are important contributors in Minsk and Kiev, particularly at the high end of the distribution and to the 90[th] percentile daily average $PM_{2.5}$ in Warsaw. The energy sector is a large contributor across the distribution in Warsaw and at the 90[th] percentile value in Budapest. In Minsk, the energy sector is more important at the low end of the distribution. SOA contributions are important at the high end of the distribution in Minsk and Warsaw.

Boundary condition contributions to summertime $PM_{2.5}$ are less important at the selected cities in western Europe. Non-road transport and SOA are the largest contributors in London and Amsterdam, particularly at the high end of the distribution. In Berlin, SOA contributions are important across the $PM_{2.5}$ distribution, while non-road transport contributions are important at the 75[th] and 90[th] percentile values. The contributions of boundary conditions are small to negligible and SOA contributions are high in all 4 selected Nordic cities. Non-road transport contributions are important in Copenhagen and Oslo, particularly
at the high ends of the distribution. The energy sector is the highest contributor at the 90[th] percentile value in Stockholm, but less important at the lower quantiles. Energy sector contributions are small but non-negligible in Helsinki.

Initial condition contributions are also included in the boundary condition contributions shown in Table 7 and Figure 9. At the end of the one week spin-up period on August 1, estimated initial condition contributions to 24-hour average
summertime $PM_{2.5}$ concentrations at 6 cities ranged from 6% at Warsaw (24-hour average $PM_{2.5}$ of 14 µg/m$^3$) to 17% at Oslo (24-hour average $PM_{2.5}$ of 13 µg/m$^3$). These initial condition contributions are expected to be smaller for the monthly average $PM_{2.5}$ concentrations shown in Table 7.

### 3.3 $PM_{2.5}$ source apportionment-winter

Figure C3 in Supplemenct C shows the spatial distribution of monthly mean $PM_{2.5}$ concentrations for February 2010 across
the modeling domain. The highest $PM_{2.5}$ concentrations are again along the southern boundary of the modeling domain, but the influence of boundary conditions further inside the domain is lower than for the summertime $PM_{2.5}$ concentrations, as

shown in Table 8. High PM$_{2.5}$ concentrations are also predicted over Poland and we see from Table 8 that, from the 16 cities selected for the analysis, the highest PM$_{2.5}$ concentration (38 µg/m$^3$) is in Warsaw.

As mentioned above, Table 8 shows that boundary condition contributions to wintertime PM$_{2.5}$ concentrations in cities along the Mediterranean coastline are much lower than summertime contributions, particularly at cities in the west, such as Lisbon and Barcelona, where BC contributions are less than 5%. BC contributions are slightly higher than 10% in the eastern Mediterranean cities (Athens and Istanbul). There are some variabilities in source contributions among the 4 Mediterranean cities. In Lisbon, SOA is the single largest contributor, explaining nearly 50% of the winter month average PM$_{2.5}$. Residential combustion is the next largest contributor at 15%, followed by non-road transport at 13%. Non-road transport is the largest contributor (21%) in Barcelona, followed by SOA, on-road transport and residential combustion with comparable contributions (17% to 18%). Residential combustion is the largest contributor in both Athens (20%) and Istanbul (25%). Non-road transport is the next highest contributor in Athens while on-road transport, industry and boundary conditions are the second highest contributors (11%) in Istanbul. Energy sector contributions are more important in the eastern Mediterranean cities (9 to 10%) than in the western cities (less than 5% in Lisbon and 7% in Barcelona). Road transport contributions in Lisbon and Athens are 10% or less. Dust emissions within the modeling domain contribute 10% of the PM$_{2.5}$ in Athens.

At the 4 selected cities in central and Eastern Europe, residential combustion is the single largest contributor to wintertime PM$_{2.5}$, with contributions ranging from 29 to 38%. Boundary condition contributions are less than 5% in all 4 cities. Road transport and the energy sector are the second highest contributors in Budapest (17 to 18%) followed by agriculture at 15%. In Kiev, agriculture and the energy sector are the second highest contributors (11% to 12%) followed by on-road and non-road transport at 10% and industry at 9%. Agriculture is the second highest contributor in Minsk (16%) followed by the energy sector and on-road transport (12% to 13%). In Warsaw, the second highest contributions to wintertime PM$_{2.5}$ are from on-road transport and agriculture (16 to 17%) while the energy sector contributes 12%. Non-road and industry contributions in Warsaw are comparable and less than 10%.

There is significant variability in the source sector PM$_{2.5}$ contributions among the cities in western Europe. In London, non-road transport and SOA are the largest contributors (23%) followed by on-road transport (19%) and residential combustion (11%). Nearly 90% of the SOA concentration in London is from biogenic precursors. Note that the source attribution simulation conducted in this study only considers source categories and does not distinguish among source regions. Thus, the SOA concentration in London could be of local origin or transported. The main contributors to the biogenic SOA concentrations in London were oxidation products of monoterpenes (46%) and sesquiterpenes (11%) as well as oligomerization of oxidized compounds (27%). CAMx includes 4 pathways for monoterpene oxidation (oxidation by OH,

$O_3$, $NO_3$, or atomic oxygen) and 3 pathways (OH, $O_3$, $NO_3$) for sesquiterpenes. The importance of wintertime SOA in a number of different European countries is discussed in Aksoyoglu et al. (2011) and Crippa et al. (2013).

Contributions from agriculture and the energy sector to wintertime $PM_{2.5}$ in London are about 7%. In Amsterdam, on-road and non-transport are the largest contributors (18 to 19%), residential combustion ranks second (16%) and agriculture and SOA contribute 12 to 13%. The energy sector contributes 10% of wintertime $PM_{2.5}$ in Amsterdam, while the industry sector contributes 7%. In both Paris and Berlin, residential combustion is the largest contributor (30% and 24%, respectively). However, there are differences in the contributions of the other source sectors in these 2 cities. SOA and on-road transport contributions rank second in Paris at about 16% followed by non-road transport at 13%, and 6 to 8% contributions from agriculture and the energy and industry sectors. In Berlin, on-road transport also ranks second but the contribution of SOA is only about 6%. Agriculture (15%), the energy sector (12%), non-road transport (11%) and industry (7%) are also significant contributors to wintertime $PM_{2.5}$ in Berlin.

Table 8 shows that, for all 4 cities in the Nordic countries, the contribution of boundary conditions is less than 5%. The largest contributors in Oslo and Helsinki are residential combustion sources (47% and 33%, respectively). The non-road and on-road transport sectors have significant contributions as well in these two cities (16% and 11% in Oslo, respectively and 14% and 18% in Helsinki, respectively). SOA, the energy sector and agriculture contribute 5 to 7% and 7 to 9% of the wintertime $PM_{2.5}$ in Oslo and Helsinki, respectively. Residential combustion is also the largest contributor in Copenhagen (20%) but it is followed closely by non-road transport (19%). Road transport contributes 14% of the wintertime $PM_{2.5}$ in Copenhagen and agriculture, the energy sector and SOA contribute about 11 to 12%. Industry contributions in Copenhagen are about 6%. Road transport is the largest contributor (22%) in Stockholm but residential combustion and non-road transport are significant contributors as well with contributions of about 19% and 16%, respectively. SOA contributes 14% to wintertime $PM_{2.5}$ in Stockholm while the energy sector contributes about 10% and agriculture and industry contribute 6 to 7%.

The source apportionment results for the 16 cities across the distribution of daily average wintertime $PM_{2.5}$ concentrations are shown in Figure 10. The contributions of boundary conditions are negligible to small across the distribution in all cities. SOA contributions in Lisbon dominate other sources at the 90[th] percentile value and are also important at the 50[th] and 75[th] percentile values. Residential combustion sources are important across the entire distribution in Istanbul and are the primary contributors at the 90[th] percentile value. The industry and agriculture sectors are also important contributors to the higher levels of wintertime $PM_{2.5}$ in Istanbul. Residential combustion is an also important contributor in Athens and Barcelona, as is the non-road transport sector.

Residential combustion is the largest contributor across the distribution of wintertime $PM_{2.5}$ concentrations in the 4 selected cities in central and western Europe. Other important sectors are agriculture and road transport. Non-road transport contributions are also important in Minsk, Warsaw and Kiev, and the energy sector is important in Minsk, Budapest and Warsaw. SOA contributions are large in 3 (London, Paris, Amsterdam) of the 4 cities in western Europe, particularly at the

90th percentile levels in London and Paris. The importance of wintertime SOA in Paris is consistent with the findings of Crippa et al. (2013). Residential combustion is an important source in Paris, Amsterdam and Berlin and has non-negligible contributions in London as well. The energy sector is an important source in Berlin and, to a smaller extent, in Amsterdam. The road and non-road transport sectors are also important contributors in all 4 cities.

Residential combustion is an important source sector in the 4 Nordic cities, particularly Oslo, where contributions from this sector dominate over the entire distribution of wintertime $PM_{2.5}$, SOA is the largest contributor at the 75th percentile level in Stockholm and is also an important contributor in Copenhagen. Both transport sectors are important in all 4 cities, with the non-road transport sector contributions being larger than the road transport contributions in Copenhagen and Oslo.

Initial condition contributions are also included in the boundary condition contributions shown in Table 8 and Figure 10. At the end of the one week spin-up period on February 1, estimated initial condition contributions to 24-hour average wintertime $PM_{2.5}$ concentrations at 6 cities ranged from less than 1% at many cities to 5% at Oslo. These initial condition contributions are expected to be negligible for the monthly average $PM_{2.5}$ concentrations shown in Table 8.

## 4 Discussion

The source attribution analysis results show that long-range transport of ozone from beyond Europe has a strong influence on summertime ozone in August 2010 over most of Europe. The background summertime ozone contribution, simulated by the boundary condition tracer in the OSAT simulation, is about 26 to 34% in southern Europe and 20 to 30% in central and eastern Europe. The boundary condition contributions in western Europe are larger, ranging from about 30 to 60%. In the Nordic cities, BC contributions range from about 20% in Stockholm to 40% in Oslo. Wintertime ozone levels are below the

EU standard and dominated by boundary conditions (60% to over 90%). The contribution of intercontinental transport (from North America and, to a smaller extent, from Asia) to ozone levels in Europe has been studied extensively through data analysis and modeling (e.g., Parrish et al., 1993; Wild and Akimoto, 2001; Lelieveld et al., 2002; Li et al., 2002; Naja et al., 2003; Trickl et al., 2003; Derwent et al., 2004; 2008; Auvray and Bey, 2005; Fehsenfeld et al., 2006; Guerova et al., 2006; Richards et al., 2013).

Summertime ozone contributions from biogenic emissions range from about 10% to 30%. At the cities selected for the analysis, the largest biogenic contribution of 33% is in Kiev, while the lowest contribution of 8% is in London. For anthropogenic emission sectors, the combined transportation sector (on-road and non-road transport) contributions range from 30 to 40% in cities along the Mediterranean coastline, cities in central and eastern Europe, and cities in northern Europe. In western Europe, the combined transport sector has a contribution of 20 to 30%. Contributions from the on-road transport sector are generally higher than those from the non-road transport sector, except for a few cities. The two transport sector contributions are comparable (within 3%) in Barcelona, Istanbul, London, and Oslo. Non-road transport contributions are slightly higher than on-road contributions in Oslo and Helsinki. These results for summertime ozone concentrations are qualitatively consistent with those of Tagaris et al. (2015) who found that the on-road transport sector was the largest overall anthropogenic source sector contributing to July 2006 ozone concentrations in Europe with non-road transport contributions ranking second. Pouliot et al. (2015) noted that emissions from on-road transport in Europe decreased from 2006 to 2009 while emissions from shipping increased. This explains some of the higher contributions of non-road transport to ozone concentrations in some cities that were calculated in our study.

The largest contributions of the energy sector were in central and Eastern Europe (9% to 17%) and in the Nordic cities (5% to 13%). The power sector was also identified as a major contributor in Europe in the study by Brandt et al. (2013). Industry contributions to summertime ozone were important for the Mediterranean cities and cities in central and eastern Europe, with contributions ranging from 5% to 9%.

For summertime ozone, the total contribution from sources that cannot be controlled within Europe (i.e., the boundary conditions and biogenic emissions) ranges from 39% to 69%. The largest non-controllable contributions are 69% in Paris and 64% in London where the H1MDA8 city center ozone concentrations are 44 ppb and 41 ppb, respectively, well below the 60 ppb threshold. However, lower ozone levels are not necessarily associated with higher non-controllable contributions, or vice-versa. For example, the H1MDA8 ozone concentration in Copenhagen is 44 ppb with anthropogenic sources contributing nearly 60%. The highest H1MDA8 ozone concentrations among the selected cities are predicted in Istanbul (73 ppb) and Kiev (70 ppb), and the non-controllable contributions are 50% and 54%, respectively.

Boundary conditions constitute a large fraction (40 to 50%) of the August 2010 average $PM_{2.5}$ concentrations in the Mediterranean cities. The influence of boundary conditions decreases from southern to northern Europe. This decreasing south-to-north gradient suggests that the Mediterranean cities were influenced by long-range transport of dust emissions from North Africa. These results are qualitatively consistent with numerous studies on the transport of Saharan dust and its contributions to PM levels in the Mediterranean Basin and other parts of Europe (e.g., Querol et al., 2001; 2004; 2009; Lyamani et al., 2005; Escudero et al., 2005; 2007a; 2007b; Vanderstraeten et al., 2008; Marconi et al., 2014; Duchi et al.,

2016). In contrast, there is an increasing south-to-north gradient in contributions of SOA (organic $PM_{2.5}$ formed in the atmosphere from precursor VOC species) to summertime $PM_{2.5}$ levels. Modeled SOA in Europe and North America is primarily associated with biogenic emissions (e.g., Sartelet et al., 2012). The contributions of SOA to summer PM range from 8 to 15% in the Mediterranean cities to 23 to 31% in the Nordic cities.

The anthropogenic source sector contributions to summertime average $PM_{2.5}$ vary with region. The important anthropogenic sectors in summer are the transport sector (both on-road and non-road), the energy sector, the industry sector, and agriculture. These sectors were also shown to be important for annual $PM_{2.5}$ in the EMEP (2009) study. The contributions of other anthropogenic source sectors to the mean monthly $PM_{2.5}$ are generally less than 10%, with the exception of the solvent
and product use sector, which has a contribution of over 10% in Amsterdam.

The source attribution results for wintertime $PM_{2.5}$ are significantly different from the summertime results. The contributions of boundary conditions are generally less than 5% with the exception of the eastern Mediterranean cities of Athens and Istanbul, where the BC contributions are 12 and 11%, respectively. SOA contributions are small (less than 10%) to moderate
(about 20%) at most locations, except in Lisbon, where the SOA contribution is nearly 50%.

The important anthropogenic sectors for wintertime $PM_{2.5}$ are residential combustion, the combined transport sector (on-road and non-road), the energy sector, and agriculture, again qualitatively consistent with the EMEP (2009) results. Residential combustion contributions in winter are much higher than in summer and range from over 10% in London to nearly 50% in
Oslo. Residential combustion is the largest contributor in 11 of the 16 cities studied in this work. Higher winter contributions from this sector are consistent with residential wood burning for heating in winter (e.g., Denier van der Gon et al., 2015; Crilley et al., 2015), particularly in northern Europe (e.g., Krecl et al., 2008). As shown in Supplemental Section A, primary $PM_{2.5}$ emissions from residential combustion are a factor of 10 higher in winter than in summer.

Our model results are subject to limitations in model formulation and input data. Model performance evaluations presented here and by others, such as AQMEII Phase 3 contributors (see Solazzo et al., 2016), can suggest where modeling uncertainties exist and how they can influence source contributions. Important sources of uncertainty include anthropogenic emission inventories, biogenic emissions, dust emissions, sea salt emissions, boundary conditions, meteorology, model formulation (e.g., SOA treatment). These uncertainties influence model performance as well as the source attribution
analysis. A detailed uncertainty analysis using sensitivity studies would provide more insight on the linkage between model performance and the source attribution analysis. Although such an analysis was not conducted as part of this study, it is useful to discuss how uncertainties in inputs and model formulation can introduce uncertainties in the source attribution results. For example, when differences between modeled and observed concentrations are mostly driven by meteorology we

may expect, as a first approximation, that the relative source contributions are reasonable even though the absolute contributions are not well captured. In contrast, discrepancies related to emissions, boundary conditions or model processes can be expected to bias both the absolute and relative contributions of specific sources. Uncertainties in boundary conditions, NOx emissions, and biogenic emissions are important for both $O_3$ and $PM_{2.5}$, uncertainties in SOA formation algorithms and

5 dust emissions are important for $PM_{2.5}$. For example, model underestimation for $PM_{2.5}$ in summer could be due to underestimation of OA caused by missing emission categories (e.g., intermediate VOC) and/or biased inventories (e.g., uncertain biogenic emissions) and/or biased model SOA schemes and these errors would influence the calculated source contributions. Quantifying source contributions can help assess when uncertainties are influential, keeping in mind that errors that underestimate impacts from a specific source may be less obvious than overestimation. The performance

evaluation for summertime ozone showed overestimation at many cities, particularly in Budapest (fractional bias of 35%). The source attribution analysis showed that boundary conditions had a significant contribution to summertime ozone in many cities, including a large contribution in Budapest (29%), suggesting that boundary condition contributions may be overstated leading to the overestimation bias.

The study presented here provides useful information on the contributions of sources that can be controlled (anthropogenic sources within Europe) versus non-controllable sources, such as boundary conditions and biogenic emissions, This information can be used as part of the decision making process (along with economic, political and societal considerations) by policy makers in efforts to improve air quality.

*Competing interests.* The authors declare that they have no conflict of interest.

*Acknowledgments.* This study was supported by Coordinating Research Council Atmospheric Impacts Committee (CRC Project A-102). We gratefully acknowledge the contribution of various groups from the AQMEII Phase 3 project for the databases used in this work.

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

**Table 1. SNAP sector classification of anthropogenic emissions.**

| Sector Number | Description |
|---|---|
| 1 | Energy industries (e.g., power generation and refineries) |
| 2 | Non-industrial (residential) combustion |
| 34 | Industry[*] |
| 5 | Extraction and distribution of fossil fuels |
| 6 | Solvent and other product use |
| 7 | Road transport  (includes exhaust, evaporative, tire/brake/road wear) |
| 8 | Non-road transport (includes rail, aircraft, shipping, construction equipment) |
| 9 | Waste treatment |
| 10 | Agriculture |

[*]**Sector 34 combines "industrial combustion" (SNAP 3) with "industrial processes" (SNAP 4) to mitigate inconsistent classification of sources to sector 3 or 4 (see Kuenen et al., 2014)**

5    **Table 2. CAMx model performance metrics at All (AB) and Rural (RB) background Airbase sites. Metrics are computed for daily mean concentrations for calendar year 2010.**

|  | $SO_2$ [ppb] | | $NO_2$ [ppb] | | $O_3$ [ppb] | | $O_x$ [ppb] | | $PM_{10}$ [µg/m³] | | $PM_{2.5}$ [µg/m³] | |
|---|---|---|---|---|---|---|---|---|---|---|---|---|
| **Parameter** | AB | RB | AB | RB | AB | RB | AB | RB | AB | RB | AB | RB |
| **# Observations** | 550113 | 90446 | 954709 | 141241 | 646965 | 144139 | 561059 | 108438 | 842896 | 115022 | 267121 | 36378 |
| **Observed Mean** | 2.3 | 1.6 | 14.0 | 6.9 | 26.3 | 29.2 | 36.7 | 34.6 | 27.8 | 21.7 | 17.5 | 14.5 |
| **Modeled Mean** | 1.2 | 0.9 | 6.1 | 4.9 | 31.8 | 31.6 | 36.8 | 35.8 | 22.6 | 21.8 | 14.0 | 13.9 |
| **Observed S.D.** | 4.6 | 2.2 | 10.1 | 6.1 | 11.3 | 11.1 | 10.3 | 10.1 | 22.0 | 17.0 | 15.9 | 13.5 |
| **Modeled S.D.** | 1.4 | 1.0 | 4.8 | 4.2 | 12.0 | 12.2 | 10.6 | 10.7 | 13.8 | 13.5 | 9.2 | 9.2 |
| **Mean BIAS** | -1.1 | -0.6 | -7.9 | -2.0 | 5.6 | 2.4 | 0.0 | 1.1 | -5.3 | 0.1 | -3.4 | -0.6 |
| **NMB (%)** | -47.9 | -40.9 | -56.3 | -29.4 | 21.1 | 8.2 | 0.0 | 3.3 | -18.9 | 0.4 | -19.7 | -4.2 |
| **Mean Error** | 1.7 | 1.1 | 8.4 | 3.3 | 8.6 | 7.4 | 6.8 | 6.6 | 14.4 | 11.9 | 8.3 | 7.1 |
| **NME (%)** | 72.1 | 69.2 | 60.0 | 47.8 | 32.7 | 25.2 | 18.5 | 19.1 | 51.7 | 55.0 | 47.3 | 49.3 |
| **FB (%)** | -45.4 | -37.6 | -73.1 | -29.3 | 19.7 | 6.4 | -0.3 | 2.9 | -15.4 | 3.5 | -13.7 | 4.2 |
| **FE (%)** | 81.9 | 76.5 | 81.6 | 55.6 | 33.5 | 28.2 | 19.5 | 20.0 | 53.1 | 52.6 | 49.8 | 50.1 |
| **Correlation** | 0.24 | 0.32 | 0.52 | 0.59 | 0.66 | 0.69 | 0.64 | 0.67 | 0.28 | 0.30 | 0.48 | 0.52 |
| **RMSE** | 4.6 | 2.2 | 11.7 | 5.4 | 11.1 | 9.5 | 8.9 | 8.5 | 23.1 | 18.3 | 14.5 | 11.8 |
| **IoA** | 0.3 | 0.5 | 0.6 | 0.7 | 0.8 | 0.8 | 0.8 | 0.8 | 0.5 | 0.5 | 0.6 | 0.7 |

**Table 3.** Summary of CAMx model performance evaluated at background Airbase sites belonging to the selected cities. Statistics are computed for daily mean $O_3$ concentrations over the summer season (July $1^{st}$ – September $30^{th}$).

| City | # Obs. | Obs. Mean | Model Mean | Obs. S.D. | Model S.D. | Mean Bias | NMB (%) | Mean Error | NME (%) | FB (%) | FE (%) | Corr. | RMSE | IoA |
|---|---|---|---|---|---|---|---|---|---|---|---|---|---|---|
| Amsterdam | 327 | 18.7 | 19.4 | 7.6 | 6.5 | 0.7 | 3.9 | 4.0 | 21.3 | 6.1 | 23.0 | 0.75 | 5.1 | 0.86 |
| Budapest | 341 | 27.2 | 38.3 | 8.3 | 10.4 | 11.1 | 40.9 | 11.4 | 42.1 | 35.0 | 36.2 | 0.70 | 13.4 | 0.62 |
| Helsinki | 265 | 27.0 | 28.7 | 10.3 | 7.8 | 1.7 | 6.3 | 5.3 | 19.4 | 10.2 | 21.4 | 0.78 | 6.6 | 0.86 |
| Oslo | 178 | 23.3 | 23.7 | 7.4 | 6.7 | 0.4 | 1.5 | 5.0 | 21.4 | 2.5 | 23.4 | 0.63 | 6.1 | 0.79 |
| Athens | 816 | 39.9 | 39.3 | 13.1 | 5.5 | -0.6 | -1.4 | 9.6 | 24.2 | 3.8 | 26.3 | 0.42 | 11.9 | 0.52 |
| Barcelona | 1769 | 29.6 | 34.8 | 8.0 | 6.9 | 5.2 | 17.5 | 7.7 | 26.1 | 18.0 | 25.5 | 0.44 | 9.5 | 0.60 |
| Berlin | 735 | 28.3 | 31.7 | 12.8 | 10.9 | 3.4 | 12.1 | 5.7 | 20.0 | 14.9 | 22.4 | 0.88 | 6.9 | 0.91 |
| Copenhagen | 178 | 20.2 | 24.6 | 7.3 | 6.6 | 4.4 | 21.7 | 5.3 | 26.1 | 21.8 | 26.1 | 0.77 | 6.4 | 0.80 |
| Lisbon | 179 | 34.4 | 35.4 | 10.0 | 6.8 | 1.0 | 2.9 | 5.0 | 14.7 | 5.3 | 15.1 | 0.79 | 6.3 | 0.85 |
| London | 428 | 17.4 | 20.1 | 7.0 | 5.7 | 2.7 | 15.3 | 4.9 | 28.1 | 16.8 | 29.2 | 0.55 | 6.7 | 0.72 |
| Paris | 1517 | 25.4 | 25.7 | 8.0 | 6.6 | 0.3 | 1.2 | 3.7 | 14.6 | 3.0 | 15.5 | 0.81 | 4.7 | 0.89 |
| Warsaw | 451 | 24.6 | 30.3 | 9.6 | 9.5 | 5.8 | 23.5 | 6.6 | 26.7 | 23.8 | 26.3 | 0.80 | 8.3 | 0.82 |

**Table 4.** Summary of CAMx model performance evaluated at background Airbase sites belonging to the selected cities. Statistics are computed for daily mean PM$_{2.5}$ concentrations over the summer season (July 1$^{st}$ – September 30$^{th}$).

| City | # Obs. | Obs. Mean | Model Mean | Obs. S.D. | Model S.D. | Mean Bias | NMB (%) | Mean Error | NME (%) | FB (%) | FE (%) | Corr. | RMSE | IoA |
|---|---|---|---|---|---|---|---|---|---|---|---|---|---|---|
| Amsterdam | 204 | 11.5 | 13.6 | 5.3 | 4.8 | 2.1 | 18.3 | 4.6 | 40.3 | 18.9 | 37.9 | 0.38 | 6.0 | 0.62 |
| Helsinki | 541 | 9.6 | 8.9 | 6.9 | 6.4 | -0.6 | -6.8 | 4.4 | 46.1 | -6.4 | 47.7 | 0.51 | 6.6 | 0.70 |
| Oslo | 533 | 8.1 | 7.7 | 2.9 | 3.6 | -0.4 | -4.9 | 2.8 | 34.3 | -11.0 | 37.5 | 0.43 | 3.5 | 0.65 |
| Athens | 163 | 25.2 | 14.0 | 8.0 | 6.4 | -11.2 | -44.4 | 11.5 | 45.7 | -60.0 | 61.6 | 0.57 | 13.1 | 0.55 |
| Barcelona | 630 | 13.8 | 11.1 | 5.4 | 5.0 | -2.7 | -19.6 | 5.4 | 39.3 | -22.5 | 41.7 | 0.22 | 7.1 | 0.51 |
| Berlin | 537 | 13.0 | 8.7 | 5.0 | 3.3 | -4.3 | -33.3 | 5.3 | 40.9 | -37.9 | 48.1 | 0.30 | 6.7 | 0.53 |
| Copenhagen | 172 | 10.3 | 10.4 | 4.0 | 4.6 | 0.1 | 1.3 | 3.7 | 36.2 | -2.3 | 36.5 | 0.38 | 4.8 | 0.62 |
| Lisbon | 172 | 11.3 | 10.1 | 5.7 | 8.6 | -1.3 | -11.1 | 5.4 | 47.8 | -11.2 | 50.8 | 0.48 | 7.8 | 0.66 |
| London | 560 | 10.7 | 10.5 | 4.0 | 4.3 | -0.2 | -1.9 | 3.9 | 36.5 | -3.8 | 35.7 | 0.24 | 5.1 | 0.54 |
| Paris | 430 | 11.0 | 12.6 | 4.7 | 5.0 | 1.6 | 14.7 | 3.9 | 35.0 | 15.5 | 32.6 | 0.55 | 4.8 | 0.72 |
| Warsaw | 276 | 19.9 | 9.7 | 9.0 | 5.3 | -10.2 | -51.3 | 11.2 | 56.3 | -67.1 | 73.1 | 0.29 | 13.6 | 0.49 |
| Stockholm | 482 | 7.2 | 6.8 | 4.0 | 2.9 | -0.4 | -5.0 | 3.1 | 43.5 | -1.2 | 42.0 | 0.25 | 4.4 | 0.50 |

**Table 5. Summary of CAMx model performance evaluated at background Airbase sites belonging to the selected cities. Statistics are computed for daily mean PM$_{2.5}$ concentrations over the winter season (January 1$^{st}$ – March 31$^{st}$).**

| City | # Obs. | Obs. Mean | Model Mean | Obs. S.D. | Model S.D. | Mean Bias | NMB (%) | Mean Error | NME (%) | FB (%) | FE (%) | Corr. | RMSE | IoA |
|---|---|---|---|---|---|---|---|---|---|---|---|---|---|---|
| Amsterdam | 260 | 25.6 | 23.6 | 19.6 | 11.3 | -2.0 | -7.7 | 10.3 | 40.3 | 3.1 | 38.6 | 0.56 | 16.3 | 0.67 |
| Budapest | 104 | 28.0 | 21.9 | 15.7 | 10.0 | -6.1 | -21.8 | 10.9 | 38.9 | -17.4 | 41.5 | 0.38 | 16.2 | 0.59 |
| Helsinki | 507 | 11.8 | 18.1 | 6.4 | 8.3 | 6.3 | 53.3 | 8.1 | 68.7 | 42.8 | 55.0 | 0.34 | 10.6 | 0.51 |
| Oslo | 532 | 15.9 | 18.7 | 9.6 | 11.7 | 2.8 | 17.8 | 11.2 | 70.7 | 14.9 | 59.2 | -0.14 | 16.4 | 0.21 |
| Athens | 212 | 19.3 | 12.9 | 11.4 | 9.4 | -6.4 | -33.3 | 9.2 | 47.5 | -36.4 | 50.3 | 0.34 | 13.7 | 0.56 |
| Barcelona | 404 | 19.3 | 12.5 | 9.8 | 4.8 | -6.9 | -35.6 | 8.8 | 45.3 | -36.1 | 50.2 | 0.31 | 11.7 | 0.51 |
| Berlin | 431 | 34.7 | 24.1 | 26.1 | 11.1 | -10.5 | -30.4 | 16.9 | 48.7 | -18.1 | 48.3 | 0.45 | 25.6 | 0.56 |
| Copenhagen | 164 | 13.9 | 20.9 | 7.9 | 10.6 | 7.1 | 51.2 | 9.7 | 70.0 | 36.2 | 54.2 | 0.42 | 12.5 | 0.57 |
| Lisbon | 167 | 8.6 | 10.4 | 4.8 | 4.8 | 1.8 | 21.3 | 5.0 | 58.2 | 22.4 | 53.2 | 0.14 | 6.6 | 0.45 |
| London | 499 | 18.0 | 20.6 | 10.2 | 9.0 | 2.6 | 14.4 | 7.2 | 39.8 | 16.5 | 37.3 | 0.58 | 9.2 | 0.74 |
| Paris | 323 | 24.3 | 25.4 | 14.3 | 11.8 | 1.0 | 4.2 | 10.4 | 42.6 | 10.0 | 43.1 | 0.50 | 13.3 | 0.68 |
| Warsaw | 278 | 42.4 | 26.3 | 21.7 | 13.2 | -16.1 | -38.0 | 18.4 | 43.4 | -43.4 | 51.7 | 0.51 | 24.7 | 0.60 |
| Stockholm | 491 | 9.6 | 14.2 | 6.1 | 7.8 | 4.6 | 48.4 | 6.8 | 71.3 | 34.6 | 54.8 | 0.38 | 9.1 | 0.56 |

**Table 6. Sectors contributing 5% or more to summertime H1MDA8 ozone concentrations. Sector contributions in % are shown in parentheses.**

| City (ppb) | Sector[*] Contributions (%) | | | | | |
|---|---|---|---|---|---|---|
| Barcelona (58) | BC (28) | SNAP 7 (21) | SNAP 8 (18) | Biogenic (15) | SNAP 34 (7) | SNAP 1 (5) |
| Lisbon (61) | BC (34) | SNAP 7 (20) | Biogenic (19) | SNAP 8 (11) | SNAP 34 (6) | SNAP 1 (6) |
| Athens (69) | BC (26) | SNAP 7 (24) | SNAP 8 (16) | Biogenic (15) | SNAP 1 (8) | SNAP 34 (6) |
| Istanbul (73) | BC (26) | Biogenic (24) | SNAP 7 (15) | SNAP 8 (13) | SNAP 34 (9) | SNAP 1 (8) |
| Minsk (58) | BC (25) | Biogenic (23) | SNAP 7 (19) | SNAP 1 (15) | SNAP 8 (10) | -- |
| Budapest (63) | SNAP 7 (35) | BC (29) | SNAP 1 (11) | Biogenic (10) | SNAP 8 (7) | SNAP 34 (5) |
| Warsaw (66) | BC (28) | SNAP 7 (24) | SNAP 1 (17) | Biogenic (14) | SNAP 8 (7) | SNAP 34 (7) |
| Kiev (70) | Biogenic (33) | BC (21) | SNAP 7 (18) | SNAP 8 (10) | SNAP 1 (9) | SNAP 34 (6) |
| London (41) | BC (56) | SNAP 8 (12) | SNAP 7 (11) | Biogenic (8) | -- | -- |
| Paris (44) | BC (59) | SNAP 7 (13) | Biogenic (10) | SNAP 8 (6) | SNAP 6 (6) | -- |
| Amsterdam (51) | BC (29) | Biogenic (21) | SNAP 7 (19) | SNAP 6 (10) | SNAP 8 (8) | SNAP 1 (6) |
| Berlin (56) | BC (46) | SNAP 7 (17) | SNAP 1 (13) | Biogenic (11) | SNAP 8 (6) | -- |
| Copenhagen (44) | BC (29) | SNAP 7 (23) | SNAP 8 (14) | SNAP 1 (13) | Biogenic (12) | SNAP 34 (5) |
| Oslo (50) | BC (41) | Biogenic (20) | SNAP 8 (14) | SNAP 7 (12) | -- | -- |
| Helsinki (50) | BC (31) | SNAP 8 (21) | SNAP 7 (17) | SNAP 1 (13) | Biogenic (13) | -- |
| Stockholm (57) | SNAP 7 (24) | BC (21) | SNAP 8 (18) | Biogenic (18) | SNAP 1 (12) | -- |

[*]See Table 1 for anthropogenic (SNAP) sector descriptions

**Table 7. Sectors contributing 5% or more to summertime monthly mean PM$_{2.5}$ concentrations. Sector contributions in % are shown in parentheses.**

| City (µg/m³) | Sector* Contributions (%) | | | | | |
|---|---|---|---|---|---|---|
| Lisbon (11) | BC (45) | SNAP 8 (18) | SOA (15) | SNAP 34 (6) | SNAP 7 (5) | -- | -- |
| Barcelona (12) | BC (40) | SNAP 8 (19) | SOA (11) | SNAP 7 (10) | SNAP 34 (5) | -- | -- |
| Athens (16) | BC (38) | SNAP 1 (15) | SNAP 8 (10) | SOA (9) | SNAP 10 (8) | SNAP 7 (7) | SNAP 34 (6) |
| Istanbul (17) | BC (49) | SNAP 34 (11) | SOA (8) | SNAP 1 (8) | SNAP 10 (7) | SNAP 8 (6) | -- |
| Budapest (10) | SNAP 1 (23) | BC (23) | SNAP 10 (13) | SOA (13) | SNAP 34 (9) | SNAP 7 (8) | SNAP 8 (5) |
| Warsaw (13) | SNAP 1 (24) | BC (21) | SOA (13) | SNAP 10 (12) | SNAP 7 (10) | SNAP 8 (8) | SNAP 34 (8) |
| Minsk (13) | BC (27) | SOA (18) | SNAP 10 (14) | SNAP 1 (14) | SNAP 7 (8) | SNAP 8 (7) | SNAP 34 (7) |
| Kiev (13) | BC (37) | SOA (17) | SNAP 10 (12) | SNAP 1 (9) | SNAP 8 (9) | SNAP 34 (6) | -- |
| Berlin (8) | SOA (19) | SNAP 1 (15) | SNAP 8 (14) | BC (14) | SNAP 7 (12) | SNAP 34 (10) | SNAP 10 (8) |
| London (10) | SOA (32) | SNAP 8 (23) | SNAP 7 (13) | BC (12) | SNAP 1 (7) | SNAP 34 (5) | -- |
| Paris (11) | SOA (18) | SNAP 8 (16) | SNAP 10 (14) | BC (14) | SNAP 7 (13) | SNAP 34 (8) | SNAP 1 (8) |
| Amsterdam (13) | SNAP 8 (28) | SOA (23) | SNAP 7 (13) | SNAP 1 (9) | BC (9) | SNAP 34 (6) | SNAP 10 (6) |
| Oslo (8) | SNAP 8 (25) | SOA (25) | SNAP 2 (11) | BC (10) | SNAP 7 (9) | SNAP 1 (7) | SNAP 34 (6) |
| Stockholm (8) | SOA (31) | SNAP 8 (15) | BC (15) | SNAP 1 (12) | SNAP 7 (10) | SNAP 34 (7) | -- |
| Helsinki (8) | SOA (31) | BC (15) | SNAP 8 (15) | SNAP 7 (13) | SNAP 1 (10) | SNAP 34 (5) | -- |
| Copenhagen (11) | SNAP 8 (26) | SOA (23) | BC (11) | SNAP 7 (10) | SNAP 1 (10) | SNAP 10 (7) | SNAP 34 (6) |

*See Table 1 for anthropogenic (SNAP) sector descriptions

**Table 8. Sectors contributing 5% or more to wintertime monthly mean PM$_{2.5}$ concentrations. Sector contributions in % are shown in parentheses.**

| City (µg/m$^3$) | Sector[*] Contributions (%) | | | | | | |
|---|---|---|---|---|---|---|---|
| Lisbon (13) | SOA (47) | SNAP 2 (15) | SNAP 8 (13) | SNAP 7 (7) | SNAP 34 (6) | -- | -- | -- |
| Barcelona (13) | SNAP 8 (21) | SOA (18) | SNAP 7 (18) | SNAP 2 (17) | SNAP 10 (7) | SNAP 1 (7) | SNAP 34 (7) | -- |
| Athens (15) | SNAP 2 (20) | SNAP 8 (17) | SOA (13) | BC (12) | Dust (10) | SNAP 7 (10) | SNAP 1 (9) | -- |
| Istanbul (26) | SNAP 2 (25) | SNAP 7 (11) | BC (11) | SNAP 34 (11) | SNAP 1 (10) | SNAP 8 (10) | SNAP 10 (9) | SOA (6) |
| Budapest (30) | SNAP 2 (29) | SNAP 7 (18) | SNAP 1 (17) | SNAP 10 (15) | SNAP 8 (7) | SNAP 34 (7) | -- | -- |
| Minsk (30) | SNAP 2 (33) | SNAP 10 (16) | SNAP 1 (13) | SNAP 7 (12) | SNAP 8 (10) | SNAP 34 (7) | -- | -- |
| Kiev (31) | SNAP 2 (37) | SNAP 10 (12) | SNAP 1 (11) | SNAP 8 (10) | SNAP 7 (10) | SNAP 34 (9) | -- | -- |
| Warsaw (38) | SNAP 2 (34) | SNAP 7 (17) | SNAP 10 (16) | SNAP 1 (12) | SNAP 8 (7) | SNAP 34 (6) | -- | -- |
| London (21) | SNAP 8 (23) | SOA (23) | SNAP 7 (19) | SNAP 2 (11) | SNAP 10 (7) | SNAP 1 (6) | -- | -- |
| Paris (25) | SNAP 2 (30) | SOA (16) | SNAP 7 (16) | SNAP 8 (13) | SNAP 10 (8) | SNAP 1 (6) | SNAP 34 (6) | -- |
| Amsterdam (26) | SNAP 7 (19) | SNAP 8 (18) | SNAP 2 (16) | SNAP 10 (13) | SOA (12) | SNAP 1 (10) | SNAP 34 (7) | -- |
| Berlin (32) | SNAP 2 (24) | SNAP 7 (18) | SNAP 10 (15) | SNAP 1 (12) | SNAP 8 (11) | SNAP 34 (7) | SOA (6) | -- |
| Stockholm (17) | SNAP 7 (22) | SNAP 2 (19) | SNAP 8 (16) | SOA (14) | SNAP 1 (10) | SNAP 10 (7) | SNAP 34 (6) | -- |
| Oslo (19) | SNAP 2 (47) | SNAP 8 (16) | SNAP 7 (11) | SOA (7) | SNAP 1 (6) | SNAP 10 (5) | -- | -- |
| Helsinki (21) | SNAP 2 (33) | SNAP 7 (18) | SNAP 8 (14) | SNAP 1 (9) | SOA (9) | SNAP 10 (7) | SNAP 34 (5) | -- |
| Copenhagen (24) | SNAP 2 (20) | SNAP 8 (19) | SNAP 7 (14) | SNAP 10 (12) | SNAP 1 (12) | SOA (11) | SNAP 34 (6) | -- |

[*]See Table 1 for anthropogenic (SNAP) sector descriptions

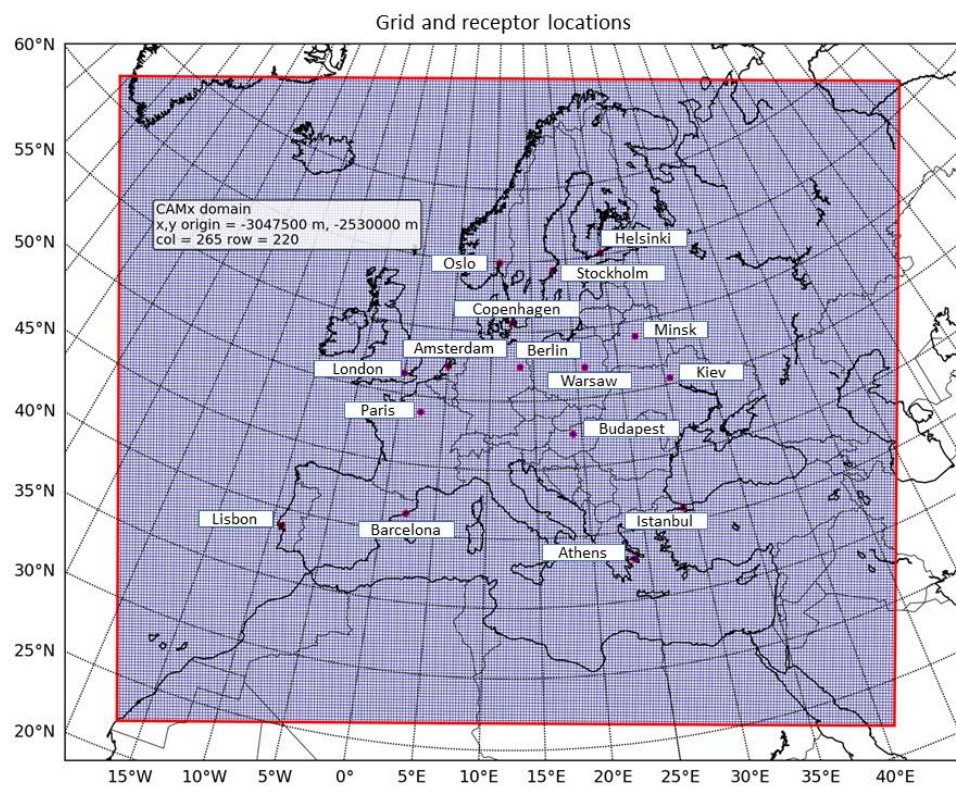

**Figure 1. CAMx modeling domain with 270 by 225 grid cells at 23 km horizontal grid resolution. The figure also shows the 16 cities considered for the source attribution analysis.**

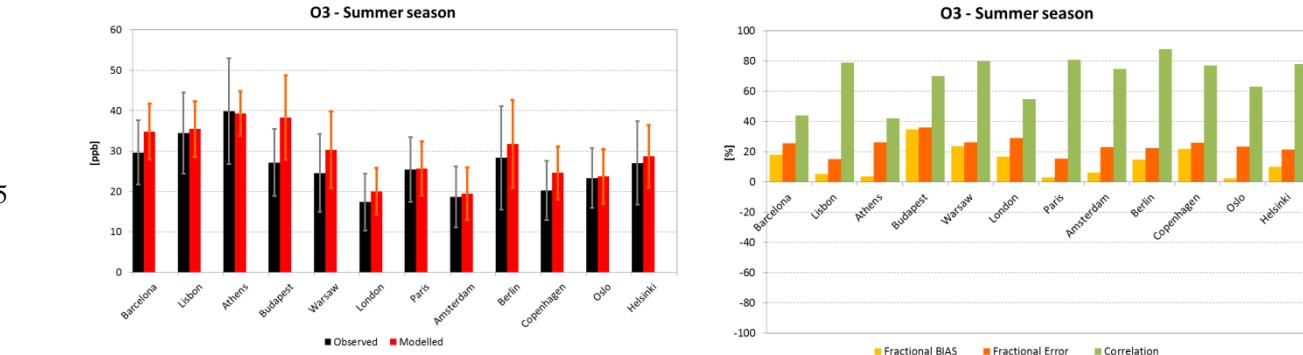

Figure 2. Evaluation of CAMx model performance for $O_3$ at selected cities for July 1 to September 30, 2010. Left panel compares the observed (black) and modeled (red) seasonal mean concentrations. Bars show the corresponding observed (grey) and modeled (orange) standard deviations. Right panel shows the seasonal Fractional Bias (orange), Fractional Error (red) and Correlation (green) computed for each city.

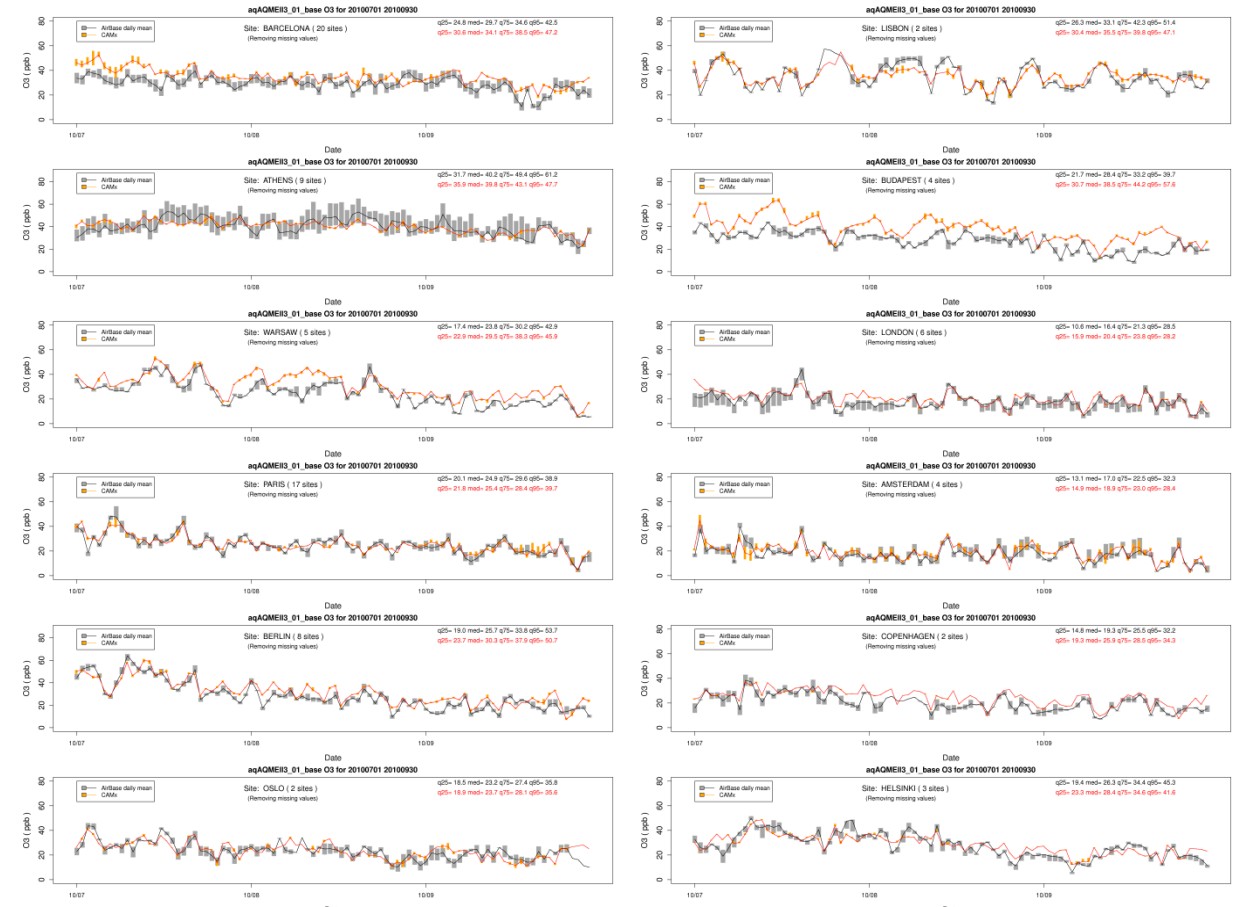

**Figure 3. Time series of the box and whisker plots for the distribution of the observed (black/grey) and computed (red/orange) daily concentrations of O₃ at Background Airbase sites in the selected cities for July 1ˢᵗ – September 30ᵗʰ 2010. Bars show the interquartile range, lines the median values. Values for the 25th, 50th, 75th, and 95th quantiles are also reported for each city.**

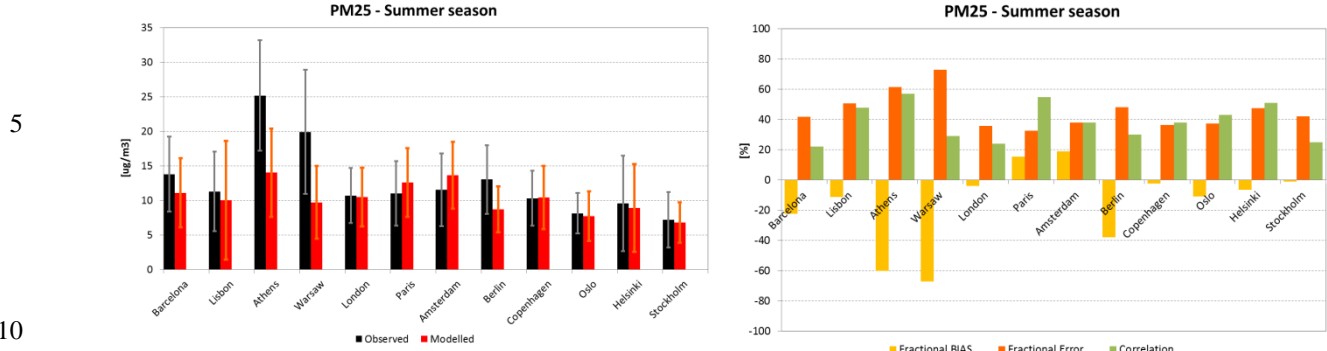

**Figure 4. Evaluation of CAMx model performance for PM$_{2.5}$ at selected cities for July 1 to September 30, 2010. Left panel compares the observed (black) and modeled (red) seasonal mean concentrations. Bars show the corresponding observed (grey) and modeled (orange) standard deviation. Right panel shows the seasonal Fractional Bias (orange), Fractional Error (red) and Correlation (green) computed for each city.**

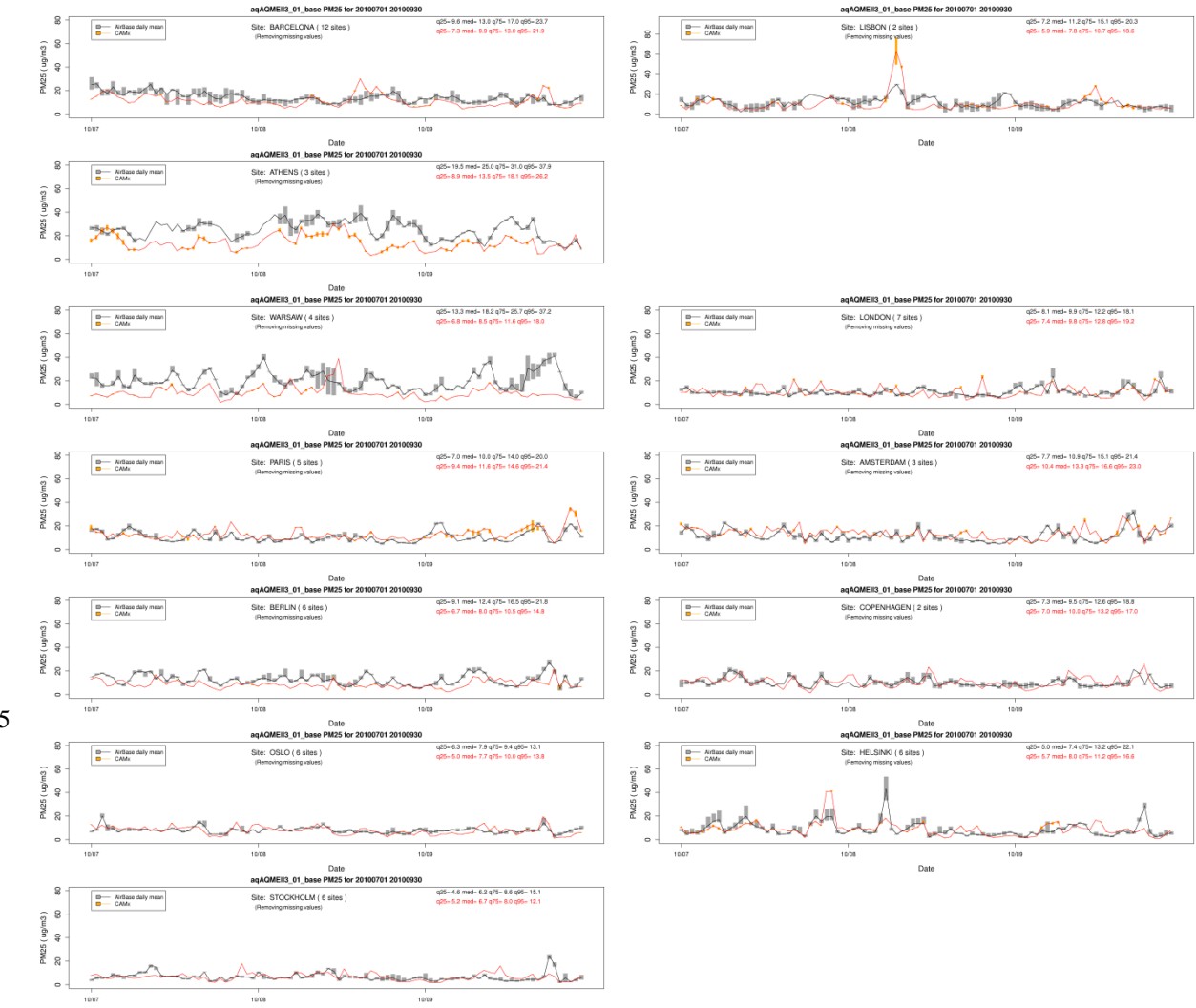

**Figure 5. Time series of the box and whisker plots for the distribution of the observed (black/grey) and computed (red/orange) daily concentrations of PM$_{2.5}$ at Background Airbase sites in the selected cities for July 1$^{st}$ – September 30$^{th}$ 2010. Bars show the interquartile range, lines the median values. Values for the 25th, 50th, 75th, and 95th quantiles are also reported for each city.**

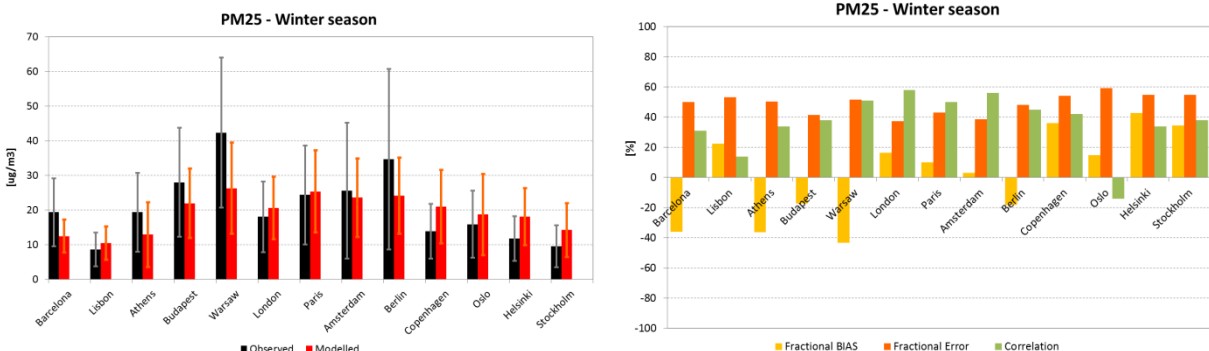

**Figure 6. Evaluation of CAMx model performance for PM$_{2.5}$ at selected cities for January 1 to March 31, 2010. Left panel compares the observed (black) and modeled (red) seasonal mean concentrations. Bars show the corresponding observed (grey) and modeled (orange) standard deviation. Right panel shows the seasonal Fractional Bias (orange), Fractional Error (red) and**
5   **Correlation (green) computed for each city.**

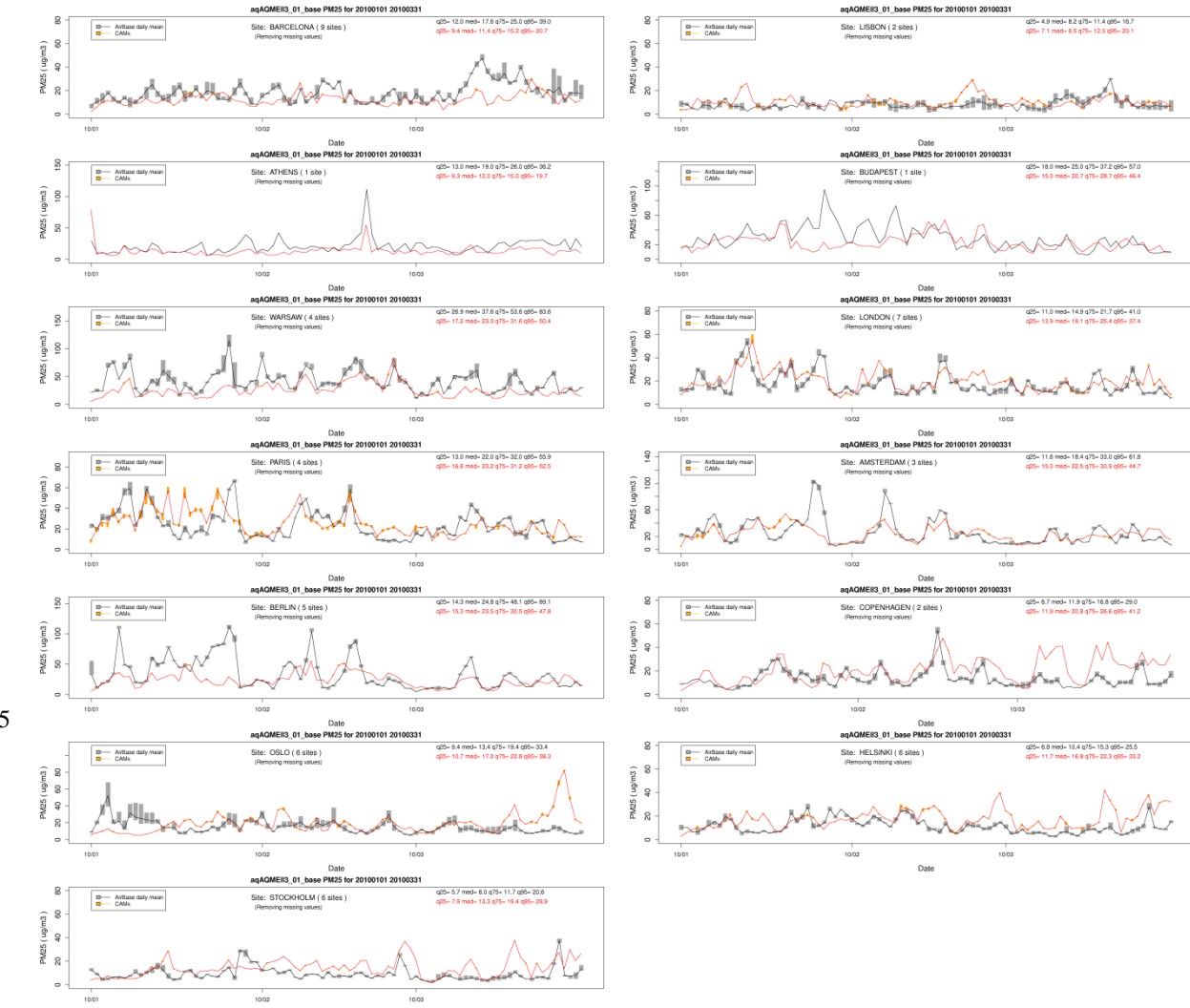

**Figure 7.** Time series of the box and whisker plots for the distribution of the observed (black/grey) and computed (red/orange) daily concentrations of PM$_{2.5}$ at Background Airbase sites in the selected cities for January 1$^{st}$ – March 31$^{st}$ 2010. Bars show the interquartile range, lines the median values. Values for the 25th, 50th, 75th, and 95th quantiles are also reported for each city.

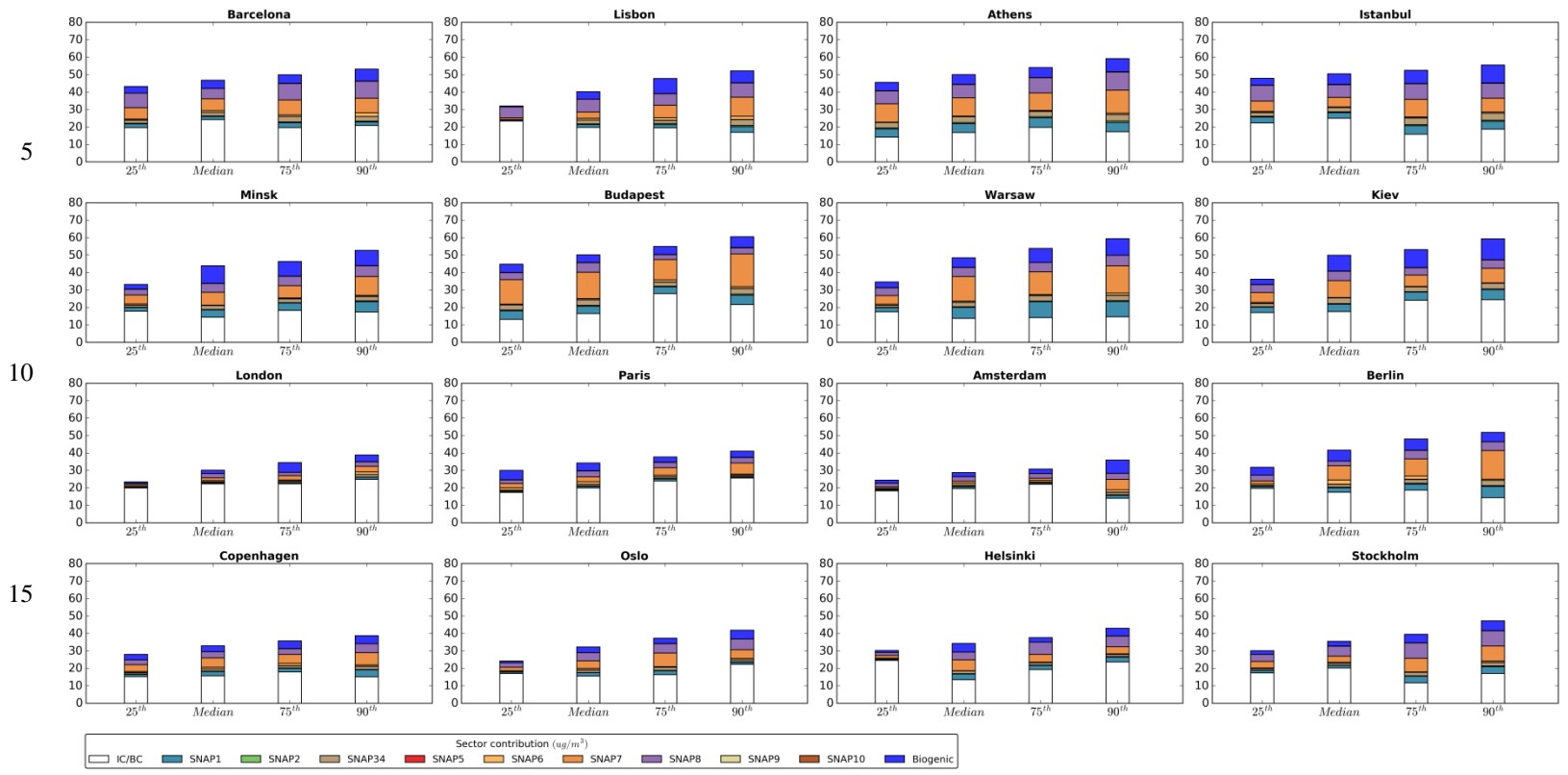

**Figure 8. Source attribution results for the distribution of summertime MDA8 ozone concentrations for the 16 cities.**

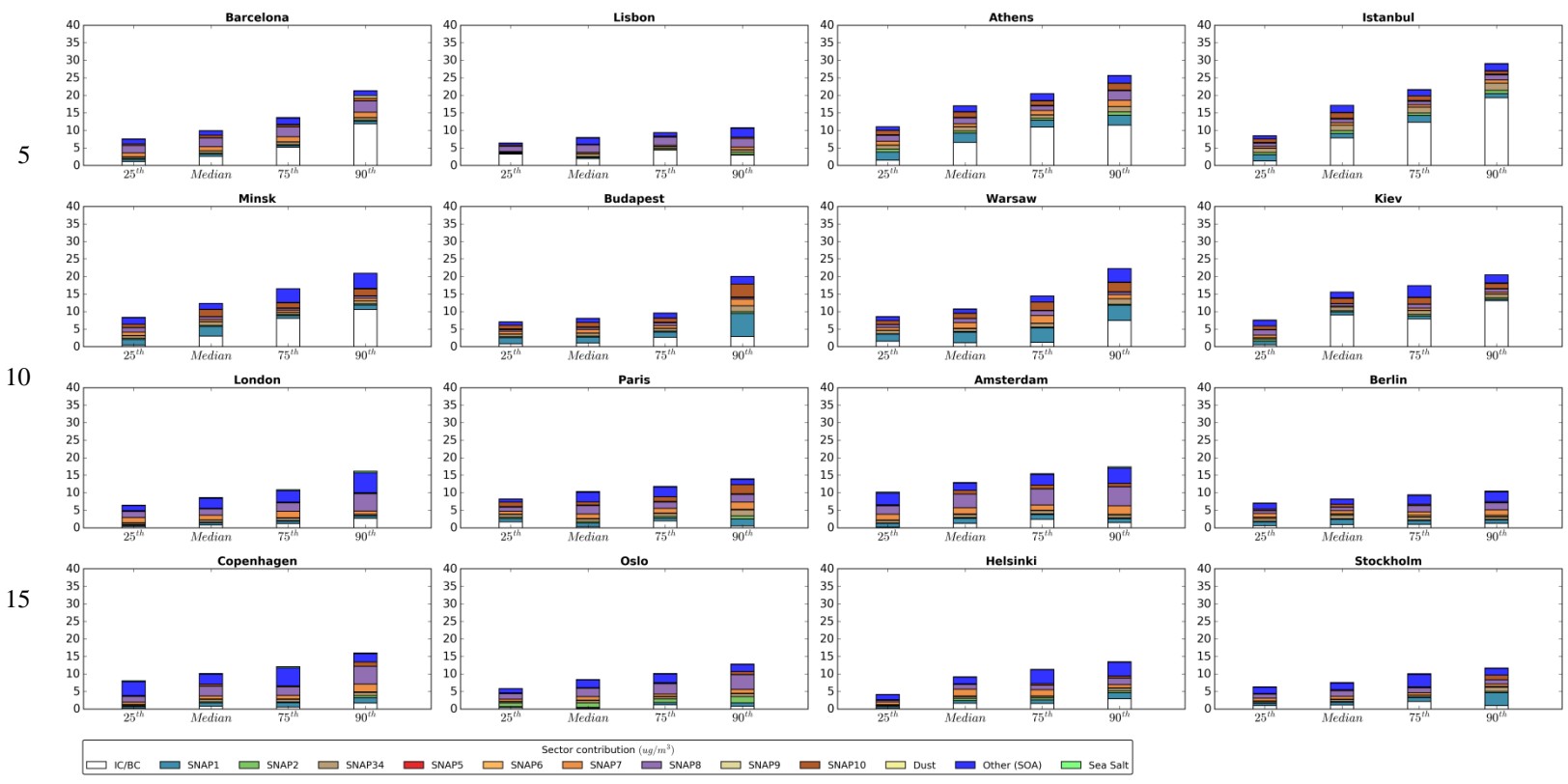

**Figure 9. Source attribution results for the distribution of summertime daily PM$_{2.5}$ concentrations for the 16 cities.**

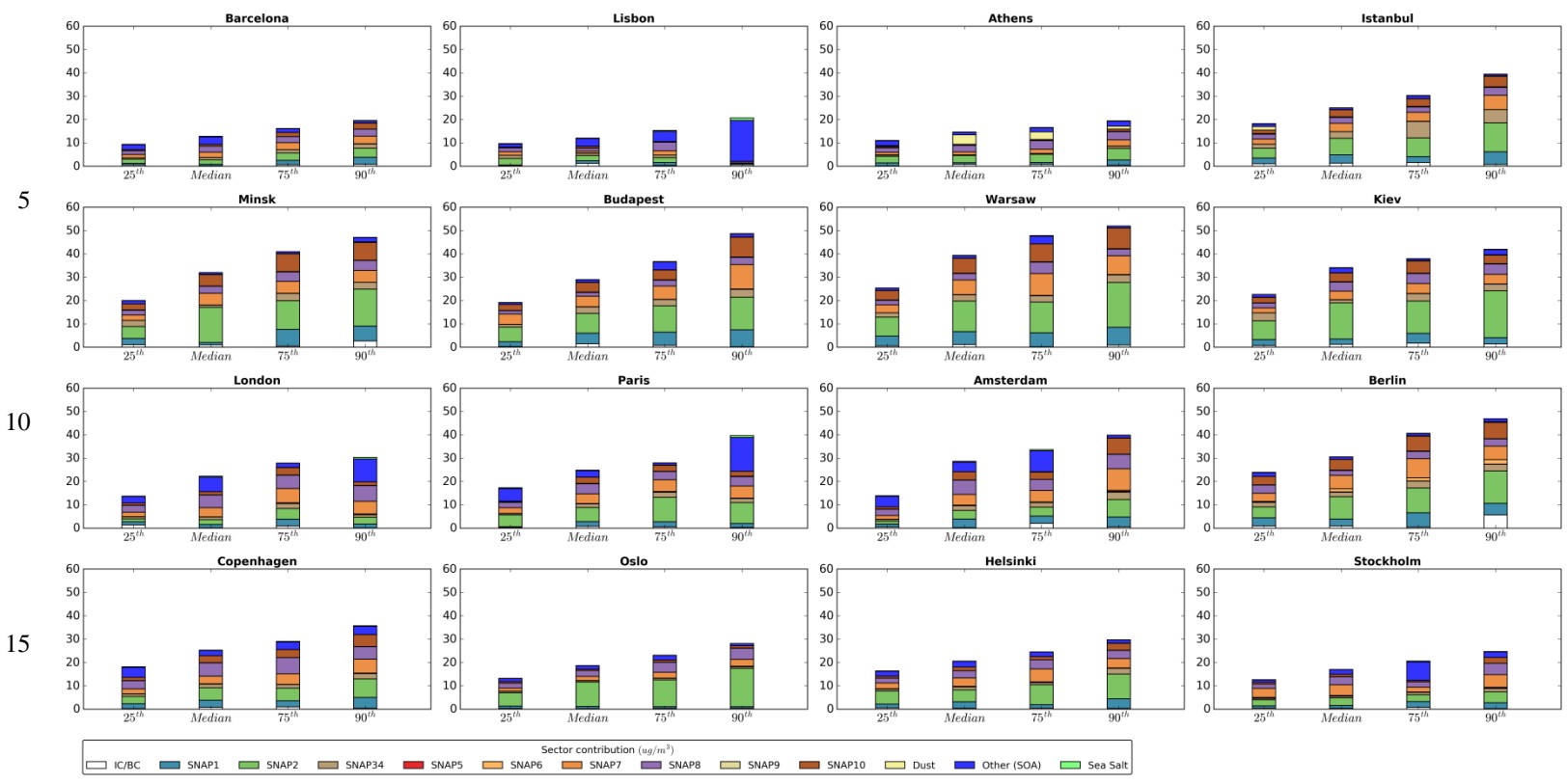

Figure 10. Source attribution results for the distribution of wintertime daily PM$_{2.5}$ concentrations for the 16 cities.