# Peer review of "Source-sector contributions to European ozone and fine PM in 2010 using AQMEII modeling data"

_Atmospheric Chemistry and Physics, 2016_

## Referee Comment (RC1) · Anonymous Referee #1 · 21 Dec 2016

This is a well-written manuscript and its subject matter fits within the scope of the special issue. The organization is clear and the figures and tables are of good quality. The amount of data produced by the source apportionment simulations is impressive. That said, in my opinion the analysis currently presented in the manuscript does not fully utilize the information available from these simulations and should be expanded to provide a more comprehensive picture of source contributions to modeled pollutant concentrations over Europe. Moreover, the manuscript should attempt to make a stronger connection between the model evaluation results and the source attribution calculations. This would not only provide the reader with a sense of the confidence in the source apportionment results, but in turn the source apportionment results may

provide information on potential sources of model errors. I encourage the authors to consider the following three main comments in revising their manuscript.

1. The model evaluation results should be made more relevant to the source apportionment results. To this end, evaluation results should to presented for the 16 cities used in the source apportionment analysis, using the monitor(s) available in the grid cell(s) used to define each city in the source apportionment analysis. Furthermore, evaluation results should be presented for the concentration metrics that are used in the source apportionment analysis. This means that for ozone, evaluation results should be presented for summer H1MDA8 and either the H8MDA8 or some other metric related to the top 8 summer MDA8 values (see comment below) while for PM2.5, summer and winter seasonal mean results should be presented separately. If the source apportionment results are expanded for additional percentiles (as recommended in the next comment), this should be done for the model performance analysis as well.

2. The source apportionment results should be presented for a wider range of conditions. For ozone, the analysis should at a minimum include analysis of the eighth highest MDA8 value or some other metric related to the top 8 MD8 values to make the source apportion results relevant to the top 24 or top 25 values over a three-year period for which the European standard is defined. However, from a process level perspective, I would strongly recommend to show the source apportionment results across the entire distribution of daily MDA8 ozone or daily mean PM2.5 values for a given season. One way to do this might be to show stacked bar charts of sector/source contributions for the 5th, 10th, .. 95th percentiles of simulated concentrations, with the first stacked bar showing the contributions at the 5th percentile, the second stacked bar showing the contributions at the 10th percentile, etc. Such plots could then be used to discuss how the relative importance of different sources and factors changes across the distribution which would provide a fuller picture of the modeled pollutant burden across a range of conditions in the different cities. In addition to the discussion of peak ozone already included in the manuscript, the discussion of contributions to low-to-midrange ozone

would be of particular interest for this special issue from a background and long-range transport perspective.

3. Finally, the manuscript should be expanded to provide a more thorough analysis and discussion on how model performance for extreme ozone values and across percentiles affects the interpretation of the source apportionment results beyond the fairly high-level statements on page 13, lines 1 – 14.

Specific comments:

Page 1, second-to-last line: remove "comma" after "(Byun and Schere, 2006)"

Page 3, lines 1-10: please also reference the work on long-range transport under Task Force for Hemispheric Transport of Air Pollution (TF-HTAP) that is a key driver for the third phase of AQMEII.

Page 3, lines 23 – 24. Was any nudging performed above the PBL? Please also specify the nudging coefficients.

Page 6, line 3: Did the source apportionment analysis include initial conditions as a factor? If so, what was the value of that factor at the end of the one week spin-up period?

Page 6, lines 5-7: Please provide the number of grid cell(s) selected for each city when calculating the source apportionment results.

Page 9, line 12: suggest rewording "is not to the same extent as for" as "is lower than for"

Page 10, line 8 and page 12, lines 22-23: what are the dominant pathways for SOA formation in London during winter? The 23% (roughly 4-5 ug/m3) contribution during winter is noteworthy.

---

## Referee Comment (RC2) · Anonymous Referee #2 · 23 Dec 2016

Dear Editor,

This MS presents a source apportionment analysis of ozone and PM2.5 using a modelling approach. The paper is very well written and structured, with clear conclusions for the reader. I have two main concerns, though, which should be addressed prior to publication: 1) uncertainty calculations: the MS does not present any uncertainty estimates for the source contributions. These would be highly necessary to support the robustness of the author's approach and results. 2) variability of the traffic source with regard to ozone source contributions: at present, there is no reference in the MS to the composition of the vehicle fleet (% diesel vs gasoline, % of EURO4-6 type vehicles) in the different cities under study, while this has a strong influence on the NO2/NO ratios

and the emissions of primary NO2 in the cities, which in turn strongly influences ozone formation. For example, the vehicle fleet composition is very different in Amsterdam, Barcelona, Athens or Helsinki, and this will influence the relevance of this source with regard to ozone formation. A discussion should be included regarding this in the MS.

Specific comments:

- Abstract and text: please define "boundary conditions", what do the authors refer to with this term? Are these the meteorological conditions, e.g. the boundary layer height? Or the model boundary conditions? Please clarify. - page 4, lines 17-18: "provides an indication...", how? Please describe in more detail. Why does the model perform better (Table 2) for RB sites only? - page 6, line 19: please clarify that the numbers in brackets in table 4 are the % of source contributions. Otherwise it may seem it is a number of days, in general it is confusing. page 6, line 25: "non-road transport", is this agricultural vehicles? Ships? - page 7, lines 9-18: the contribution from road transport depends on the vehicle fleer (dieselisation, EURO...). It is relevant to add a discussion on this for a comparison between the different cities. - page 7, line 12: 12% of non-road transport seems very high for London, where do these emissions come from? Especially if road transport accounts for 11%. Please review or explain this result. - page 8, line 1: Saharan dust is morly coarse (>2.5 microns). Please add refs if the impact is also observed in <2.5 particles - page 11, lines 19-20: the non-road transport source should be further explained; if it is mostly shipping emissions it will make much more sense than if it is focussing on agricultural and construction vehicles, for example. Please clarify. - Discussion: the references to "controllable" and "non-controllable" sources are interesting, and this discussion could be extended further maybe with 1 paragraph summarising the applicability of this study, e.g., based on the authors' results, what can be done to reduce ozone and PM2.5 concentrations in the cities under study? Which sources can be controlled, and what kind of reductions (quantifiable) could be achieved if these sources were targeted?

---

## Referee Comment (RC3) · Anonymous Referee #3 · 26 Dec 2016

Review to the paper

Source-sector contributions to European ozone and fine PM in 2010 using AQMEII modeling data

by P. Karamchandani, Y. Long, G. Pirovano, A. Balzarini, and G. Yarwood

In this work, the authors make use of calculations performed for year 2010 with CAMx model within the AQMEII project to make a source apportionment of ozone and PM pollution in European cities. Namely, the relative contributions from 10 SNAP source categories, and also from SOA, natural PM sources and boundary conditions to calculated concentrations of $O_3$ and $PM_{2.5}$ in February and August 2010 are estimated

for 16 cities. So-called, tagged species methods for ozone and PM (OSAT/PSAT) were used to make the source allocations and the main calculation results are presented in the paper.

The work presented in the paper is very relevant in order to identify the main sources of health damaging pollution in different urban locations, which could contribute to design more optimal emission reduction strategies. The paper is well written, with the material being presented clearly and neatly.

The results definitely carry interesting information, but unfortunately the authors limite most of the paper to describing what a reader him/herself can easily see in the allocation tables. It'd be interesting to learn how the main presented results differ from other source allocations exercises, what added information is obtained in this work, and also a short discussion on the implication of the results could be recommended. Furthermore, some discussion regarding uncertainties in the apportionment results due to (spatially variable) inaccuracies in model calculations could be relevant here.

Specific comments:

1. Model setup. As the boundary conditions (BCs) appear a major source, could the authors specify what species were included in the BCs and provide a short discussion on their accuracy. How the cities were defined in model results: by one model grid cell, or depending on the city size?

2. Model evaluation. I think it'd me more relevant to show the model performance for the selected months, February and August (especially since the authors indicate some model problems with modelled temporal evolution). I'd recommend to move annual mean evaluation to Supplement, and rather include the evaluation of model performance for those two months in the main part.

It'd also be relevant to include information regarding the model accuracy with respect to PM components (in particular SOA and mineral dust) that has effect the accuracy of

source apportionment.

3. Results and Discussion. To my opinion, Sections 3 and 4 give a too detailed narrative about the results in Tables 3-5. Basically those sections are describing the same numbers, just from two different angles. As mentioned above, I'd suggest more discussion on the credibility and uncertainty of the results due to modelling inaccuracies (for instance, regarding BCs and SOA, which are often calculated to be major sources). The result that BCs are the one main source of summertime PM2.5 in Minsk and Kiev is surprising to me. Could the author specify which PM components from beyond Europe that contribute so much to PM pollution in those cities.

It'd also be very interesting to see comparison of the obtained results with other studies, including those which applied a "zero-out" approach (Google search shows at least one such Chapter 5 in http://emep.int/publ/reports/2009/status_report_1_2009.pdf).

Another relevant issue for discussion is how representative the presented source apportionment result are given quite significant effects of meteorology, especially for relatively short periods (could the weather conditions characterized as "typical" in February and August 2010)?

Further I wonder how well the PSAT method cope with the situations when PM pollution is due to more than one SNAP source? For instance PM episodes were registered last years in France (Paris), Germany, the UK etc. caused by ammonia emissions from agriculture and urban traffic emissions. Tables 4 and 5 identify several cities where SNAP7 and SNAP10 are among the major contributors.

Minor comments:

p.2 line 9: typo in (Colette et al., 2014), also this reference is missing.

Table 2. Specify what correlation is shown

Figure 1. I'm not sure it's so important to include it. Or I'd suggest to show the cities considered on the map and the grid cells they are covered with. Figures 3-5: the colour

scales chosen make it impossible to spot the regions with exceedances of the critical levels; unless they are changes I cannot see much sense in showing those maps.

---

## Author Response (AR2)

We thank referee #1 (RC1) for the thoughtful review of our manuscript and the constructive comments on how it could be improved. Our responses to the comments and the resulting revisions to the manuscript are listed below.

**General Comment 1:** We agree with the referee that additional model performance evaluation (MPE) at city level and summer and winter seasons would support the analysis of SA results. For this project, observed data collected for MPE included only daily mean concentrations, therefore the additional analysis was limited to this metric, although SA results for ozone refer to the H8MDA. However, we believe such analysis can provide enough information about CAMx model performance at the selected cities and for the two seasons.

In response to the referee request, we have added a new section (Sect. 2.2.2) where MPE at city level is presented, along with new tables and figures. The previous performance evaluation for annual values is now in Sect. 2.2.1. The new MPE is conducted for both ozone and PM2.5. Two different seasons (January-March and July-September) are considered.

**General Comment 2:** As requested by the referee, we have conducted additional analyses of the source attribution results to consider a wider range of conditions. Specifically, we have looked at source contributions for the 16 cities for the 25th, 50th, 75th and 90th percentiles of daily MDA8 summertime ozone as well as daily mean summer and winter PM2.5, to supplement the previous analyses that only considered H1MDA8 ozone and monthly PM2.5. Sects. 3.1, 3.2, and 3.3 have been revised accordingly, along with new figures.

**General Comment 3:** We have updated Sect. 4 (Discussion) to discuss in more detail the relationship between model performance and source attribution results. Additional sensitivity studies and uncertainty analysis would need to be performed to provide more quantitative relationships, and such an uncertainty analysis was not part of the current study.

**Specific Comments:**

All comments have been addressed in the revised manuscript:

- 1) Instead of removing the "comma" after (Byun and Schere, 2006) as suggested by the referee, we have added another "comma" after (PGMs) at the beginning of the sentence.
- 2) The requested reference to TF-HTAP has been added in the Introduction section.
- 3) Nudging on wind speed, temperature and water vapor mixing ratio has been performed above and within the Boundary Layer, with the same nudging coefficient of 0.0003 sec-1. The text in Sect. 2.1 was modified accordingly to the Reviewer request.
- 4) The initial conditions were included as part of the boundary conditions. The abstract and text have been modified accordingly and the contribution of the initial conditions at the end of the one week spin-up is discussed in the source attribution results in Sect. 3.
- 5) The source apportionment results were conducted using horizontal bilinear interpolation over 8 grid cells around each city location. This information is now included in the manuscript in the introduction to Sect. 3.

- 6) The requested correction has been made.
- 7) Sect. 3.3 has been revised in response to the referee's request seeking more information on the pathways for wintertime SOA contributions to PM2.5 in London. Two new references have also been added as part of this revision.

We thank referee #2 (RC2) for the thoughtful review of our manuscript and the constructive comments on how it could be improved. Our responses to the comments and the resulting revisions to the manuscript are listed below.

**General Comment 1:** An uncertainty analysis was not conducted as part of the current study. However, we have expanded the Discussion Section (Sect. 4) to discuss in more detail the sources of uncertainties, and the relationship between model performance and source attribution results both of which are influenced by these uncertainties. Also, please see the response to General Comment 1 to RC1.

**General Comment 2:** The reviewer has requested information on the composition of the vehicle fleet in different cities in Europe with a discussion on how this might influence the source apportionment results. This information is not available, because the emission inventory was made available to the AQMEII participants with sources contribution grouped according the SNAP classification, but without any additional information about the car fleet or other proxies introduced in emission computation. However, the MACC-II emission inventory (Kuenen et al., 2014) that was used for this project was constructed by using the reported emission national totals by sector. Therefore, for each country a representative car fleet was used. The manuscript has been revised accordingly in Sect. 2.1 to include this information.

**Specific Comments:**

All comments have been addressed in the revised manuscript:

- 1) As requested by the referee, boundary conditions are more precisely defined in the abstract and in the text.
- 2) Section 2.2 has been revised to provide more information on why model performance at RB sites is expected to better than performance at UB sites.
- 3) The source attribution table captions have been updated to point out that the source contributions are provided in %.
- 4) A sentence on the source categories included in non-road transport has been added in Sect. 2.1.
- 5) This comment is already addressed in the response to General Comment 2 above.
- 6) The large non-road transport contributions in London are likely related to shipping activity along the Channel and the text in Sect. 3.1 has been modified accordingly.
- 7) Sect. 3.2 has been updated to cite 2 new references regarding the presence of fine PM in Saharan dust.
- 8) Sect. 3.1 has been updated to note that emissions from shipping and harbors are an important nonroad transport influence for Oslo and Helsinki, along with a citation to a new reference.
- 9) Source apportionment can provide useful information to policy makers that can be used in designing control strategies. Our study provides this information but cannot make policy decisions, which require other considerations, such as cost-benefit analysis, politics, societal impacts, etc. We have revised the discussion in Sect. 4 accordingly.

We thank referee #3 (RC3) for the thoughtful review of our manuscript and the constructive comments on how it could be improved. Our responses to the comments and the resulting revisions to the manuscript are listed below.

**General Comment:** Our initial submission already provided some qualitative comparisons with previous similar studies. More quantitative comparisons are difficult considering the large differences between what has been done in the current study versus previous source apportionment studies discussed in the Introduction Section. Nevertheless, we have attempted to include additional discussion on other similar studies, including the EMEP study suggested by the referee. Three new references are cited, and the text in Sects. 1 and 4 has been updated accordingly.

**Specific Comments:**

All comments have been addressed in the revised manuscript:

- 1) Sect. 2.1 has been revised to provide additional details on the boundary conditions and 2 new references are cited.
- The first part of this comment has been addressed in the response to General Comment 1) from RC1. The second part of this comment has been addressed in the response to General Comment 3) from RC1.
- 3) The first part of this comment has been addressed in the response to General Comment 3) from RC1. For the second part of this comment, our analysis of the boundary condition contribution to PM2.5 in Minsk and Kiev indicates that over 60% of the contribution is from primary fine crustal material. The text in Sect. 3.2 has been revised accordingly. The third part of this comment is addressed in our response to the General Comment above. For the fourth part of this comment, we have added a sentence to the first paragraph of Sect. 2.3 to provide more information on how PSAT can track multiple source contributions.

**Minor Comments:**

- The reference is to Collet et al. (2014), not Colette et al. 2014, and the reference was included in the initial submission: "Collet, S., Minoura, H., Kidokoro, T., Sonoda, Y., Kinugasa, Y., Karamchandani, P., Johnson, J., Shah, T., Jung, J., and DenBleyker, A.: Future year ozone source attribution modeling studies for the eastern and western United States, J. Air Waste Manage. Assoc., 64, 1174-1185, 2014". No change was made for this comment.
- 2) Sect. 2.2.1 has been updated to define correlation.
- 3) Figure 1: In response to the reviewer comment, the figure has been modified to show the cities considered for the source attribution analysis.
- 4) Figures 3-5: These figures have been revised as suggested by the referee, but have now been moved to the Supplemental Section to accommodate the additional analysis, tables and figures requested by RC1.

[revised manuscript text omitted]
            | 0.3               | 0.5   | 0.6             | 0.7    | 0.8    | 0.8    | 0.8               | 0.8    | 0.5     | 0.5    | 0.6       | 0.7    |

I

|                   |               |             |               |             | _           |              |                            |                      |                   |                                  |                    |              |             |             |
|-------------------|---------------|-------------|---------------|-------------|-------------|--------------|----------------------------|----------------------|-------------------|----------------------------------|--------------------|--------------|-------------|-------------|
| City       | # Obs. | Obs.        | Model
Moor | Obs.        | Model       | Mean
Biog | $\underline{\mathbf{NMB}}$ | Mean
Ermon | $\underline{NME}$ | $\frac{\mathbf{FB}}{\mathbf{P}}$ | $\frac{FE}{(9())}$ | Corr. | RMSE | IoA  |
|                   |               | Mean        | Mean          | 5.D. | 5.D. | Dias         | (70)                       | Error                | (70)              | (70)                             | (70)               |              |             |             |
| Amsterdam  | 327    | 18.7 | 19.4   | 7.6  | 6.5  | 0.7   | 3.9                 | 4.0                  | 21.3       | 6.1                       | 23.0        | 0.75  | 5.1  | 0.86        |
| Budapest   | 341    | 27.2 | 38.3   | 8.3  | 10.4 | 11.1  | 40.9                | 11.4          | 42.1       | 35.0                      | 36.2        | 0.70  | 13.4 | 0.62 |
| Helsinki   | 265    | 27.0 | 28.7   | 10.3 | 7.8  | 1.7   | 6.3                 | 5.3           | 19.4       | 10.2                      | 21.4        | 0.78  | 6.6  | 0.86 |
| Oslo       | 178    | 23.3 | 23.7   | 7.4  | 6.7  | 0.4   | 1.5                 | 5.0           | 21.4       | 2.5                       | 23.4        | 0.63  | 6.1  | 0.79 |
| Athens     | 816    | 39.9 | 39.3   | 13.1 | 5.5  | -0.6  | -1.4                | 9.6           | 24.2       | 3.8                       | 26.3        | 0.42  | 11.9 | 0.52 |
| Barcelona  | 1769   | 29.6 | 34.8   | 8.0  | 6.9  | 5.2   | 17.5                | 7.7                  | 26.1       | 18.0                      | 25.5        | 0.44  | 9.5  | 0.60 |
| Berlin     | 735    | 28.3 | 31.7   | 12.8 | 10.9 | 3.4   | 12.1                | 5.7           | 20.0       | 14.9                      | 22.4        | 0.88  | 6.9  | 0.91 |
| Copenhagen | 178    | 20.2 | 24.6   | 7.3  | 6.6  | 4.4   | 21.7                | 5.3           | 26.1       | 21.8                      | 26.1        | 0.77  | 6.4  | 0.80 |
| Lisbon     | 179    | 34.4 | 35.4   | 10.0 | 6.8  | 1.0   | 2.9                 | 5.0           | 14.7       | 5.3                       | 15.1        | 0.79  | 6.3  | 0.85 |
| London            | 428    | 17.4 | 20.1   | 7.0  | 5.7  | 2.7   | 15.3                | 4.9           | 28.1       | 16.8                      | 29.2        | 0.55  | 6.7  | 0.72 |
| Paris      | 1517   | 25.4 | 25.7   | 8.0  | 6.6  | 0.3   | 1.2                 | 3.7           | 14.6       | 3.0                       | 15.5        | 0.81  | 4.7  | 0.89 |
| Warsaw     | 451    | 24.6 | 30.3   | 9.6  | 9.5  | 5.8   | 23.5                | 6.6           | 26.7       | 23.8                      | 26.3        | 0.80  | 8.3  | 0.82 |
|                   |               |             |               |             |             |              |                            |                      |                   |                                  |                    |              |             |             |

 Table 3. Summary of CAMx model performance evaluated at background Airbase sites belonging to the selected cities. Statistics are computed for daily mean  $O_3$  concentrations over the summer season (July 1st – September 30th).

| City              | # Obs. | Obs.        | Model
Mean | Obs.               | Model              | Mean
Bios | $\frac{\mathbf{NMB}}{(0)}$ | Mean | NME  | $\frac{\mathbf{FB}}{(0)}$ | $\frac{FE}{(9())}$ | Corr. | RMSE | IoA  |
|-------------------|---------------|-------------|---------------|--------------------|--------------------|--------------|----------------------------|-------------|-------------|---------------------------|--------------------|--------------|-------------|-------------|
| Amsterdam         | 204           | 11.5        | 13.6          | 5.0.
5.3 | 5.D.
4.8 | 2.1          | 18.3                       | 4.6         | 40.3        | 18.9                      | 37.9               | 0.38         | 6.0         | 0.62        |
| Helsinki          | 541           | 9.6         | 8.9           | 6.9                | 6.4                | -0.6         | -6.8                       | 4.4         | 46.1        | -6.4                      | 47.7               | 0.51         | 6.6         | 0.70        |
| Oslo       | 533    | 8.1  | 7.7    | 2.9         | 3.6         | -0.4  | -4.9                | 2.8  | 34.3 | -11.0              | 37.5        | 0.43  | 3.5  | 0.65 |
| Athens     | 163    | 25.2 | 14.0   | 8.0         | 6.4         | -11.2 | -44.4               | 11.5 | 45.7 | -60.0              | 61.6        | 0.57  | 13.1 | 0.55 |
| Barcelona  | 630    | 13.8 | 11.1   | 5.4         | 5.0         | -2.7  | -19.6               | 5.4  | 39.3 | -22.5              | 41.7        | 0.22  | 7.1  | 0.51 |
| Berlin     | 537    | 13.0 | 8.7    | 5.0         | 3.3         | -4.3  | -33.3               | 5.3  | 40.9 | -37.9              | 48.1        | 0.30  | 6.7  | 0.53 |
| Copenhagen | 172    | 10.3 | 10.4   | 4.0         | 4.6         | 0.1   | 1.3                 | 3.7  | 36.2 | -2.3               | 36.5        | 0.38  | 4.8  | 0.62 |
| Lisbon     | 172    | 11.3 | 10.1   | 5.7         | 8.6         | -1.3  | -11.1               | 5.4  | 47.8 | -11.2              | 50.8        | 0.48  | 7.8  | 0.66 |
| London     | 560    | 10.7 | 10.5   | 4.0         | 4.3         | -0.2  | -1.9                | 3.9  | 36.5 | -3.8               | 35.7        | 0.24  | 5.1  | 0.54 |
| Paris      | 430    | 11.0 | 12.6   | 4.7         | 5.0         | 1.6   | 14.7                | 3.9  | 35.0 | 15.5               | 32.6        | 0.55  | 4.8  | 0.72 |
| Warsaw     | 276    | 19.9 | 9.7    | 9.0         | 5.3         | -10.2 | -51.3               | 11.2 | 56.3 | -67.1              | 73.1        | 0.29  | 13.6 | 0.49 |
| Stockholm  | 482    | 7.2  | 6.8    | 4.0         | 2.9         | -0.4  | -5.0                | 3.1  | 43.5 | -1.2               | 42.0        | 0.25  | 4.4  | 0.50 |

Table 4. Summary of CAMx model performance evaluated at background Airbase sites belonging to the selected cities. Statistics are computed for daily mean PM2.5 concentrations over the summer season (July  $1^{\text{st}}$  – September  $30^{\text{th}}$ ).

| mean r m2.5 con |               |                     |                      |              |               |                     |                   |                      |                   |                            |                   |              |             |             |
|-----------------------------------|---------------|---------------------|----------------------|--------------|---------------|---------------------|-------------------|----------------------|-------------------|----------------------------|-------------------|--------------|-------------|-------------|
| City                       | # Obs. | Obs.
Mean | Model
Mean | Obs.
S.D. | Model
S.D. | Mean
Bias | NMB
(%) | Mean
Error | NME
(%) | $\frac{\mathbf{FB}}{(\%)}$ | $\frac{FE}{(\%)}$ | Corr. | RMSE | IoA  |
| Amsterdam                         | 260    | 25.6                | 23.6                 | 19.6  | 11.3          | -2.0                | -7.7       | 10.3                 | 40.3              | 3.1                        | 38.6              | 0.56  | 16.3 | 0.67        |
| Budapest                   | 104    | 28.0         | 21.9          | 15.7  | 10.0   | -6.1         | -21.8      | 10.9          | 38.9       | -17.4               | 41.5       | 0.38  | 16.2 | 0.59 |
| Helsinki                   | 507    | 11.8         | 18.1          | 6.4   | 8.3    | 6.3          | 53.3       | 8.1           | 68.7       | 42.8                | 55.0       | 0.34  | 10.6 | 0.51 |
| Oslo                       | 532    | 15.9         | 18.7          | 9.6   | 11.7   | 2.8          | 17.8       | 11.2          | 70.7       | 14.9                | 59.2       | -0.14 | 16.4 | 0.21 |
| Athens                     | 212    | 19.3         | 12.9          | 11.4  | 9.4    | -6.4         | -33.3      | 9.2           | 47.5       | -36.4               | 50.3       | 0.34  | 13.7 | 0.56 |
| Barcelona                  | 404    | 19.3         | 12.5          | 9.8   | 4.8    | -6.9         | -35.6      | 8.8           | 45.3       | -36.1               | 50.2       | 0.31  | 11.7 | 0.51 |
| Berlin                     | 431    | 34.7         | 24.1          | 26.1  | 11.1   | -10.5        | -30.4      | 16.9          | 48.7       | -18.1               | 48.3       | 0.45  | 25.6 | 0.56 |
| Copenhagen                 | 164    | 13.9         | 20.9          | 7.9   | 10.6   | 7.1          | 51.2       | 9.7           | 70.0       | 36.2                | 54.2       | 0.42  | 12.5 | 0.57 |
| Lisbon                     | 167    | 8.6          | 10.4          | 4.8   | 4.8    | 1.8          | 21.3       | 5.0           | 58.2       | 22.4                | 53.2       | 0.14  | 6.6  | 0.45 |
| London                     | 499    | 18.0         | 20.6          | 10.2  | 9.0    | 2.6          | 14.4       | 7.2           | 39.8       | 16.5                | 37.3       | 0.58  | 9.2  | 0.74 |
| Paris                             | 323    | 24.3         | 25.4          | 14.3  | 11.8   | 1.0          | 4.2        | 10.4          | 42.6       | 10.0                | 43.1       | 0.50  | 13.3 | 0.68 |
| Warsaw                     | 278    | 42.4         | 26.3          | 21.7  | 13.2   | -16.1        | -38.0      | 18.4          | 43.4       | -43.4               | 51.7       | 0.51  | 24.7 | 0.60 |
| Stockholm                  | 491    | 9.6          | 14.2          | 6.1   | 7.8    | 4.6          | 48.4       | 6.8           | 71.3       | 34.6                | 54.8       | 0.38  | 9.1  | 0.56 |

[revised manuscript text omitted]

\*See Table 1 for anthropogenic (SNAP) sector descriptions